# Higher surface folding of the human premotor cortex is associated with better long-term learning capability
Marco Taubert [1,2,3] ✉, Gabriel Ziegler [3,4,5] & Nico Lehmann [1,3,6]

The capacity to learn enabled the human species to adapt to various challenging environmental conditions and pass important achievements on to the next generation. A growing body of research suggests links between neocortical folding properties and numerous aspects of human behavior, but their impact on enhanced human learning capacity remains unexplored. Here we leverage three training cohorts to demonstrate that higher levels of premotor cortical folding reliably predict individual long-term learning gains in a challenging new motor task, above and beyond initial performance differences. Individual folding-related predisposition to motor learning was found to be independent of cortical thickness and intracortical microstructure, but dependent on larger cortical surface area in premotor regions. We further show that learning-relevant features of cortical folding occurred in close spatial proximity to practice-induced structural brain plasticity. Our results suggest a link between neocortical surface folding and human behavioral adaptability.

Cortical folding is a highly complex developmental process that depends on the genotype[1] and reflects the functional organization of the cortex[2–6], with striking similarities but also numerous differences between individuals and across species[7,8]. It has been suggested that cortical folding evolved to fit a larger sheet-like cortex into a compact cranial space and to keep cortical nerve fiber connections short[9–11]. This evolutionary expansion and folding of the human neocortex, especially in associative cortices, likely enhanced the neurocomputational capacities required for complex social interaction, tool-making, and mobility[12]. Compared to cortical folding, which develops very early in prenatal and postnatal periods[13], cortical surface area increases threefold in the postnatal period and peaks at 11–12 years of age[14]. It, therefore, seems plausible to assume that differences in cortical folding in adults represent consequences of early developmental influences on behavior[13].

However, the exact role of cortical folding in behavior is still debated[15] and this topic has fascinated early neuroanatomists[16–18] and stimulates contemporary research in diverse fields such as biology, anthropology or cognitive neuroscience[12,19–21]. The dominant view is that higher levels of cortical folding are linked to improved cognitive performance both within

and across species[11,17,22,23]. Patients with certain neurodevelopmental disorders present cortical folding abnormalities and cognitive deficits[24] and cross-sectional studies in healthy populations demonstrate positive correlations between cortical morphology and behavioral performance (most frequently with parameters of 'intelligence') but with varying small to moderate effect sizes[22,25–28]. A recent prospective observational study found strong correlations between cortical folding and intra-individual changes in cognition[27], although possible differences in the extent and intensity of practice could not be taken into account. In the motor domain, previous investigations revealed performance correlations with cortical folding (at a single point in time) in developmental and clinical samples[29,30], as well as relationships between handedness with sulcation[31], speech motor recovery with gyrification[32] and expertise-related gyral differences in elderly musicians[33]. However, according to a recent review on individual difference predictors of motor learning[34], the association between cortical folding and differences in motor learning remains unexplored. We here exploit multi-cohort longitudinal data to test whether cortical folding in the motor system might form a potential predisposition for intra-individual performance gains during motor practice over several weeks.

[1]Department of Sport Science, Institute III, Faculty of Humanities, Otto von Guericke University, Zschokkestraße 32, 39104 Magdeburg, Germany. [2]Center for Behavioral and Brain Science (CBBS), Otto von Guericke University, Universitätsplatz 2, 39106 Magdeburg, Germany. [3]Collaborative Research Center 1436 Neural Resources of Cognition, Otto von Guericke University, Leipziger Str. 44, 39120 Magdeburg, Germany. [4]Germany German Center for Neurodegenerative Diseases (DZNE), Leipziger Straße 44, 39120 Magdeburg, Germany. [5]Institute of Cognitive Neurology and Dementia Research, Otto von Guericke University, Leipziger Str. 44, 39120 Magdeburg, Germany. [6]Department of Neurology, Max Planck Institute for Human Cognitive and Brain Sciences, Stephanstraße 1a, 04103 Leipzig, Germany. ✉e-mail: marco.taubert@ovgu.de

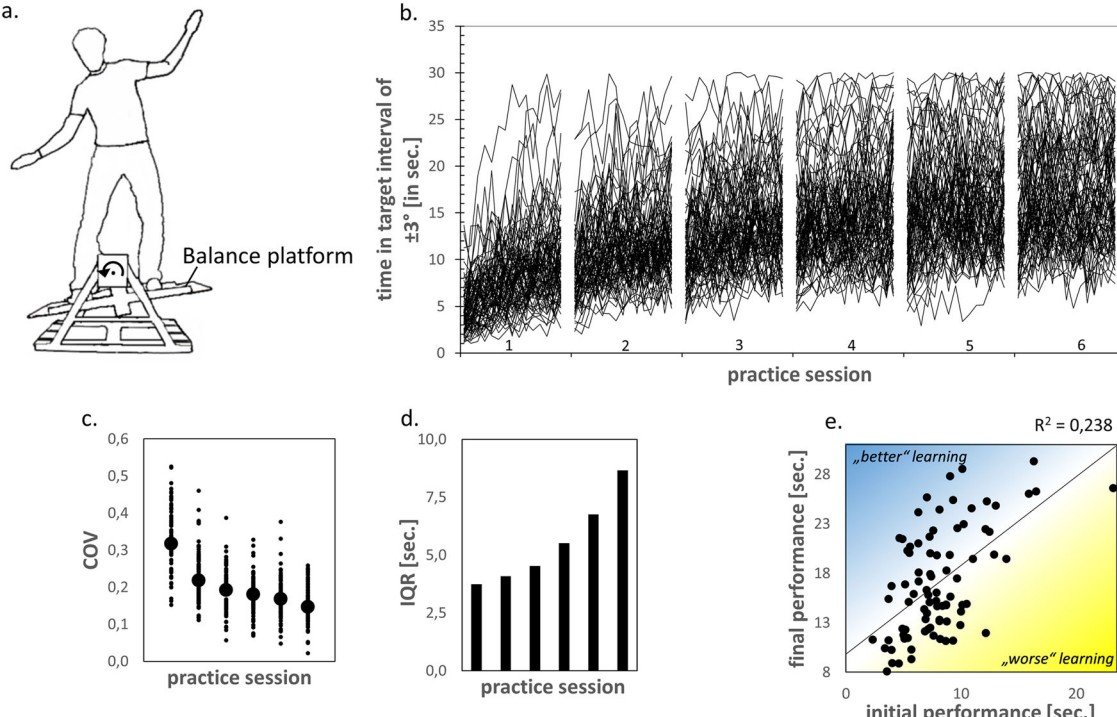

**Fig. 1 | Behavioral data.** Motor learning task, performance improvements, performance stabilization and increased inter-individual differences in motor learning over 6 practice sessions. **a** We tested motor learning of a challenging whole-body balancing task[51]. Participants were instructed to keep a seesaw-like moving stabilometer balance platform in a horizontal target interval (±3°) as long as possible during a trial length of 30 s. **b** Motor performance was measured as the time (in seconds) in which participants kept the board within the ±3° target interval in each of 15 practice trials per session (see Supplementary Video files for motor performance of participants at the beginning and end of practice). **c** Decrease in trial-to-trial variability (coefficient of variation, COV) of session-specific motor performance. **d** Increase of the interquartile range (IQR) of session-specific between-person variation in motor performance. IQR increased from 3.7 s at session 1 to 8.7 s at session 6. **e** From the first to the sixth session, participants tended to maintain their performance rank (Spearman correlation between initial and final performance, $R^2 = 0.238$, $p < 0.001$) but there were large individual differences in learning (blue/yellow: higher/lower performance than predicted from baseline).

It has been suggested that high human performance does not directly result from evolved brain features alone, but rather from an interaction between fertile learning environments and remarkable learning capacities provided by the brain[35,36]. Motor learning induces brain plasticity[37] but behavioral genetics research also indicated that practice increases the relative importance of genetic influences on performance and reduces the effects of environmental variation resulting from different prior experiences[38,39]. Research in patients indicate that the ability to perform efficient visual-based corrective movements in adulthood is highly dependent on motor experience at a very young age[40]. Therefore, learning in the human brain appears to be mediated by certain predispositions and practice-induced neural plasticity in the cortical and subcortical gray and white matter[25,28,41,42]. However, no study to date investigated whether individual differences motor learning capability are associated with relatively stable markers of cortical neuroanatomy, such as neocortical folding. Building on recent developmental studies of behaviorally relevant features of cortical shape[5] and our own work on motor learning-induced cortical plasticity[43], we hypothesize that individual variations in cortical folding does predict the individual potential to learn a new motor task and that such folding variations colocalize with learning-induced neural plasticity.

In the human brain, local geometric features of the cortical surface (e.g., cortical curvature) appear to fundamentally constrain brain function[44]. Cortical curvature can be examined in vivo using magnetic resonance imaging (MRI), providing a folding-related measure to investigate spatially-specific brain-behavior relationships[45,46]. Recent comparative[15] and experimental[4] studies indicate that, under the limited space constraints of the skull, the size, thickness and cellular composition of the cortical sheet influence the degree of cortical folding. Under equal

space conditions, a larger cortical surface area and/or a smaller cortical thickness leads to higher degrees of folding. Therefore, indices of cortical surface area, cortical thickness and intracortical microstructure enable a complementary investigation of brain-behavioral associations of cortical folding. To comprehensively characterize the link between local cortical folding and motor learning, we pursue a stepwise analysis approach. Specifically, in cortical regions with learning-relevant geometrical features (cortical curvature), we further investigate contributions of cortical surface area, cortical thickness and intracortical microstructure (assessed using myelin-sensitive magnetization transfer saturation and neurite density index).

Using data sets from previous motor learning studies, we aim to disentangle the contributions of higher cortical folding either to superior (absolute) performance or superior learning capability (steeper learning rate above and beyond initial performance differences). The joint analysis of MRI data from three separate motor learning experiments[43,47–49] allowed us to examine individual learning differences in a challenging balance task over a practice period of 4 to 6 weeks[50] (Fig. 1a). The stabilometer balance task served as a model paradigm for learning new whole-body motor coordination patterns[51]. We identified a robust positive association between higher cortical folding in premotor cortical regions and superior motor learning capability (learning rate). The effect of higher cortical folding to superior absolute performance was fully mediated by differences in learning rate. Larger cortical surface area, but not cortical thickness or cortical microstructure, contributed to the identified relation between premotor cortical folding and learning rate. A spatial overlap was identified between premotor cortical folding predispositions and learning-induced structural brain plasticity. These results suggest a link between neocortical surface folding and individual differences in learning ability.

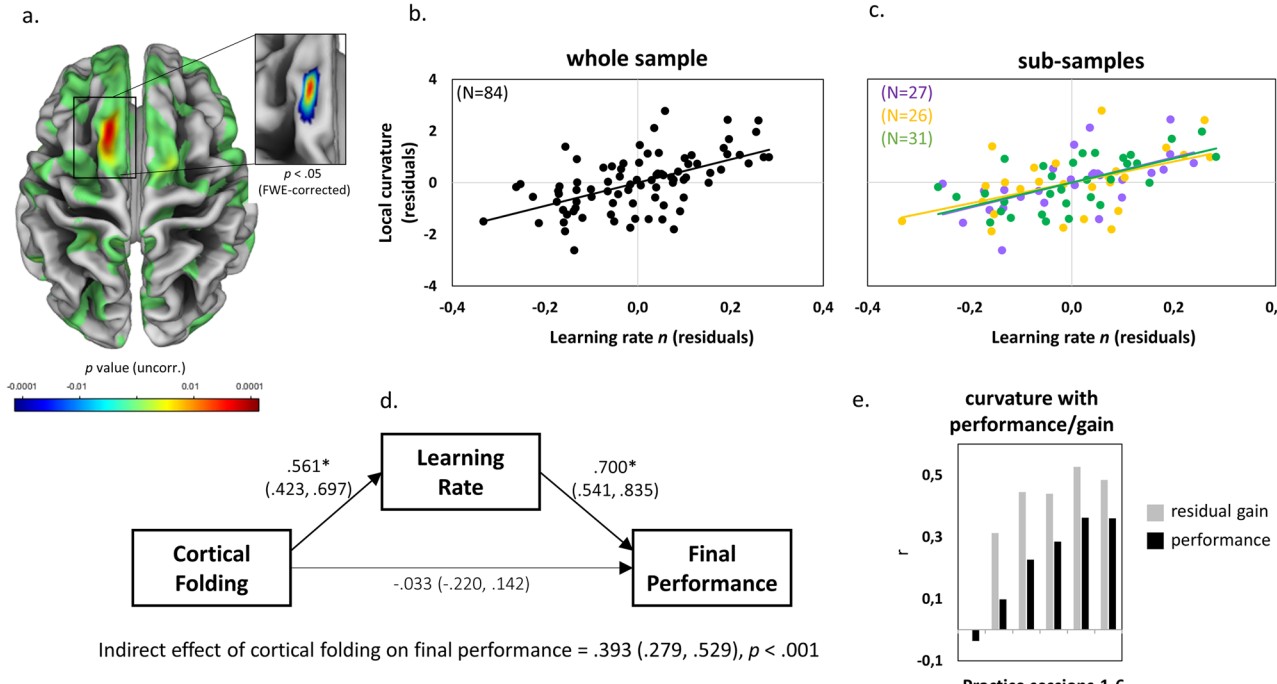

**Fig. 2 | Cortical folding is related to motor learning.** Results of whole-brain correlation of vertex-wise cortical curvature and learning rate (N = 84). **a** Uncorrected results at p < 0.001 (left) and family-wise error-corrected results at p < 0.05 (inset) were projected onto a template showing surface variations in sulcus depth. **b** Association between cortical folding (in the cluster representing the FWE-corrected effect in the original analysis [A]) and learning rate (displayed for visualization of the range of individual values only and not for inference). **c** Subsample results in the three independent learning experiments (displayed for visualization of the range of individual values only and not for inference, detailed information on sub-samples in Supplementary Table 1). **d** Structural equation model depicting relationships between cortical folding in pre-SMA/SMA (cluster from 2 A, unadjusted for a), learning rate (adjusted for a) and final performance on session 6 (unadjusted for a). Results of a separate analysis of final performance are depicted in Supplementary Fig. 8. Standardized coefficients with 95% bootstrapped confidence intervals (CI) are represented on paths. **e** Pearson correlations between cortical folding and motor performance. Gray bars represent session-specific performance controlled for initial performance in session 1 (i.e., residual gain) and black bars represent correlations with actual session-specific performance. * indicate significant paths at p < 0.05 (with CIs not including zero).

## Results

### Long-term motor practice improves performance, reduces intra-individual performance variability and enhances inter-individual performance differences

Participants practiced a whole-body balance task in six sessions evenly spaced over 4 to 6 weeks (Fig. 1a, b). Throughout the practice period motor performance increased continuously (main effect of session $F_{(5, 415)} = 202.61$, $p < 0.001$, $\eta p^2 = 0.709$) with significant performance gains across the six practice sessions (all post-hoc comparisons between time points were significant at $p < 0.001$, Bonferroni corrected for multiple comparisons). Intraindividual (trial-to-trial) variability decreased (main effect of session $F_{(5, 415)} = 109.89$, $p < 0.001$, $\eta p^2 = 0.570$, Fig. 1c) and absolute between-person performance differences (IQR) increased during practice (Fig. 1d). This shows significant inter-individual variability in motor learning (Fig. 1e). To relate the differences in motor learning to variations in cortical folding, we fitted a general power function

$$y(x) = a * x^n$$

to the session-specific mean performance scores of each participant. The intercept $a$ of the power function represents initial performance, while the exponent $n$ reflects the individual learning rate and $x$ is session. The general power function yielded an adequate fit to the individual performance data with a median coefficient of determination of $R^2 = 0.90$. In accordance with the literature[52], initial performance $a$ negatively predicted learning rate $n$ ($R^2 = 0.350$, $p < 0.001$, Supplementary Fig. 1). Therefore, we adjusted the learning rate for interindividual differences in initial performance[53] and used the term 'learning rate' for this in all subsequent analyses.

### Cortical folding is associated with inter-individual differences in motor learning

We quantified vertex-wise cortical curvature to measure local cortical folding[54]. Due to the relatively low initial image resolution ($1 \times 1 \times 1$ mm voxel size), we had to limit our analysis of the folding to the cerebral cortex. Larger cortical curvature values indicate higher degrees of local cortical folding. We tested for correlations between higher cortical curvature and steeper learning rate (adjusted learning rate $n$), higher initial performance (intercept $a$), enhanced short-term improvements within the first practice session and higher asymptotic/final performance in the last practice session 6. All analyses were adjusted for age, gender, body height, study, and total intracranial volume (see covariate correlation matrix in Supplementary Fig. 2).

We did not observe significant correlations between local cortical curvature and initial performance or short-term improvements (Supplementary Figs. 3 and 4). Instead, a steeper learning rate $n$ was positively associated with higher cortical curvature in the left pre-supplementary/supplementary motor area (pre-SMA/SMA, peak at x = −13, y = 18, z = 63, T = 5.97, whole-cortex analysis with FWE correction at $p < 0.05$, nonparametric t-statistic with 5000 permutations, see Fig. 2a, b and Supplementary Fig. 5). No dataset was excluded due to outliers in performance values. However, using a fixed threshold (two standard deviations below and above the learning rate mean), there were two outliers below participants' performance scores. A re-analysis showed that the relationship between learning rate and cortical folding was comparable in pre-SMA/SMA without these two participants ($R^2 = 0.31$ for N = 84 and $R^2 = 0.29$ for N = 82). This positive correlation was reproducible in a second MRI scan of the same participants (Supplementary Fig. 6). Approximately 30% of the variance in learning rates was explained by differences in cortical curvature in pre-

SMA/SMA ($R^2 = 0.30$, $p < 0.001$, $N = 84$). Detailed analyses of performance improvements in each of the six practice sessions revealed progressively stronger associations between cortical curvature and motor performance throughout the practice period (Fig. 2e). Lastly, the analysis of final performance revealed a non-significant trend for a positive association with cortical curvature in left pre-SMA/SMA (local maximum at x = −15, y = 20, z = 62, T = 4.40, whole-cortex analysis with FWE-corrected $p = 0.053$, nonparametric t-statistic with 5000 permutations) and a significant association in a cluster in left supramarginal gyrus (local maximum at x = −59, y = −56, z = 21, T = 4.55, whole-cortex analysis with FWE correction at $p < 0.05$, nonparametric t-statistic with 5000 permutations, see Supplementary Fig. 8). As can be seen in Fig. 2a and Supplementary Fig. 8, there was a close spatial relation between curvature correlations in left pre-SMA/SMA and learning rate as well as final performance. In order to confirm the link between cortical folding, learning rate and final performance, we used structural equation modeling (see Materials for SEM fit indices). The SEM results revealed no significant direct effect but an indirect effect of cortical folding on final performance that was fully mediated via learning rate $n$ (Fig. 2d).

### Individual folding-related predisposition to motor learning is independent of cortical thickness, but dependent on cortical surface area

At the macroscopic level, cortical folding depends on the size and thickness of the cortical sheet (surface area and cortical thickness, see ref. 15). Thus, we tested the potential contributions of cortical surface area and cortical thickness to the observed relationship between cortical folding and learning rate using SEM.

Modeling results are shown in Fig. 3a (see Materials for model fit indices). Within a larger region encompassing left pre-SMA/SMA (see Methods for ROI description), cortical surface area, but not cortical thickness, exerted an indirect effect on learning rate $n$ via folding (indirect effect of surface area on $n$: 0.54 [95% CI = 0.305, 0.749], $p < 0.001$; no indirect effect of thickness on $n$: 0.02 [95% CI = −0.076, 0.134], $p = 0.686$). We confirm a direct effect of cortical folding on learning rate $n$ within this SEM ($R^2 = 0.21$, see also Fig. 3b for Pearson correlation between cortical folding and learning rate $n$, $R^2 = 0.16$, $p < 0.001$).

Interestingly, the positive relationship between cortical folding and learning rate $n$ remained significant in a partial correlation analysis when adjusting for differences in surface area and cortical thickness ($R^2 = 0.17$, $p < 0.001$, Fig. 3c).

In order to extend the effect of premotor cortical curvature on learning rate, we used the related surface area-dependent gyrification index[55] and found a spatial pattern of positive correlations that is consistent with the premotor effects observed with the curvature-based measure (Supplementary Fig. 9). Thus, differences in cortical surface area contributed to the association between premotor folding and learning rate, but additional factors independent of cortical surface area and thickness also contributed to explain differences in learning rate.

### Cortical folding ties to learning rates independent of cortical myelination and cortical neurite density

Cross-species comparisons do suggest that highly convoluted cortices have lower neuronal densities than less convoluted cortices[56]. Also, the folding process in regions developing late during gestation (secondary and tertiary sulci) is likely to be mediated by intracortical microstructure[6] as well as biomechanical constraints[57]. In order to test whether the effect of folding on learning rate is significantly influenced by intracortical microstructure, we measured neurite density and myelin content in pre-SMA/SMA in a subsample[58] ($N = 26$; mean age 22.1 years, range 19–29 years, Fig. 4c) from which we additionally obtained quantitative and multi-shell diffusion MRI data. In this subsample, we estimated intracortical neurite density index (NDI) and myelin-sensitive magnetization transfer saturation (MT). In line with previous studies[59], we observed a positive correlation between chronological age and MT, in particularly in deep cortical gray matter, in vertex-

wise (Supplementary Fig. 10) and ROI-wise correlation analyses ($R^2 = 0.33$, $p = 0.002$, Fig. 4d). However, no significant correlations between MT and learning rate $n$ were identified, neither in ROI ($R^2$ ranged from 0.017 to 0.034, $p > 0.36$, Fig. 4e) nor in vertex-wise analyses (Supplementary Fig. 10). Partial correlations confirmed that associations between cortical folding and learning rate $n$ remained significant when controlling for variations in MT or NDI (partial $R^2$ ranged from 0.26 to 0.27, all $p < 0.009$, Fig. 4f). These results via imaging proxies in a sub-sample of our study suggest that the association between higher premotor cortical folding and steeper learning rates is less likely to be related to lower intracortical myelin content or neurite density across individuals.

### Coincident effects of cortical folding and practice-induced plasticity

Our previous study revealed structural gray matter plasticity in the pre-SMA/SMA with practice of the very same whole-body balance task[43] (Fig. 5a left). This gives us the opportunity to test the spatial coincidence of folding predispositions for learning and learning-induced plasticity. Within the cluster that showed strongest gray matter volume increases across six weeks of motor practice (Fig. 2 in ref. 43), higher cortical curvature was significantly associated with steeper learning rates (peak at x = −15, y = 18, z = 59, T = 5.64, FWE corrected $p = 0.001$, Fig. 5a right). Also, average cortical curvature in this cluster was significantly related to individual differences in learning rate ($R^2 = 0.29$, $p < 0.001$, Fig. 5b). This effect was consistent across the three sub-samples (Fig. 5c). Using SEM of 'plasticity' ROI values, we confirm the indirect effect of cortical folding on final performance (both unadjusted for $a$, Fig. 5d) mediated via learning rate $n$ (adjusted for $a$).

### Discussion

Given the complexity of mechanisms involved in the expansion and folding of the cerebral cortex, and thus its tremendous costs in terms of genetic, cellular, and histogenic evolution, the ecological advantages of cortical folding must be more than remarkable[60]. Using longitudinal training datasets, we show that human participants with higher degrees of cortical folding in a premotor area have larger performance gains (steepness of the learning rate) when learning a new whole-body balance task across several sessions of practice. These local effects of cortical folding overlapped with balance practice-induced structural plasticity in premotor areas observed in previous studies[43,58]. Our data support models that (1) attribute higher cortical folding to larger cortical surface area and (2) view individual differences in performance in adulthood as the result of (neural) predispositions that unfold in the context of new learning experiences. The results of our study support the hypothesis that higher levels of cortical folding endow individuals with enhanced adaptive capacities, but not with superior performance per se.

Interindividual differences in global and local cortical folding correlate with behavioral performance in adult humans[8,13,22,27,45,46,61–65]. Such studies usually assessed cognitive or memory performance at a single point in time – with intelligence quotient being the most commonly measured variable to date. The effect size of brain-behavior correlations varied considerably but generally suggest a positive association between higher folding and performance at a single point in time. Here, using a longitudinal measure of performance change, we report that approx. 30% of variance in learning rate can be explained by the degree of local cortical folding in premotor cortical regions (pre-supplementary/supplementary motor areas). In line with ref. 27, larger cortical surface area contributed to the identified association between cortical folding and learning rate. While the technical reproducibility (Supplementary Fig. 6) of the folding-learning association was expected because of the high stability of non-invasive markers of external brain morphology, we were surprised by the spatial overlap of positive correlations with a previously identified brain region responding with cortical plasticity through practice of the same balance task (Fig. 5c).

We report a comparably large effect (approx. 30% of explained variance) for a brain-behavioral correlation of cortical folding in adult humans

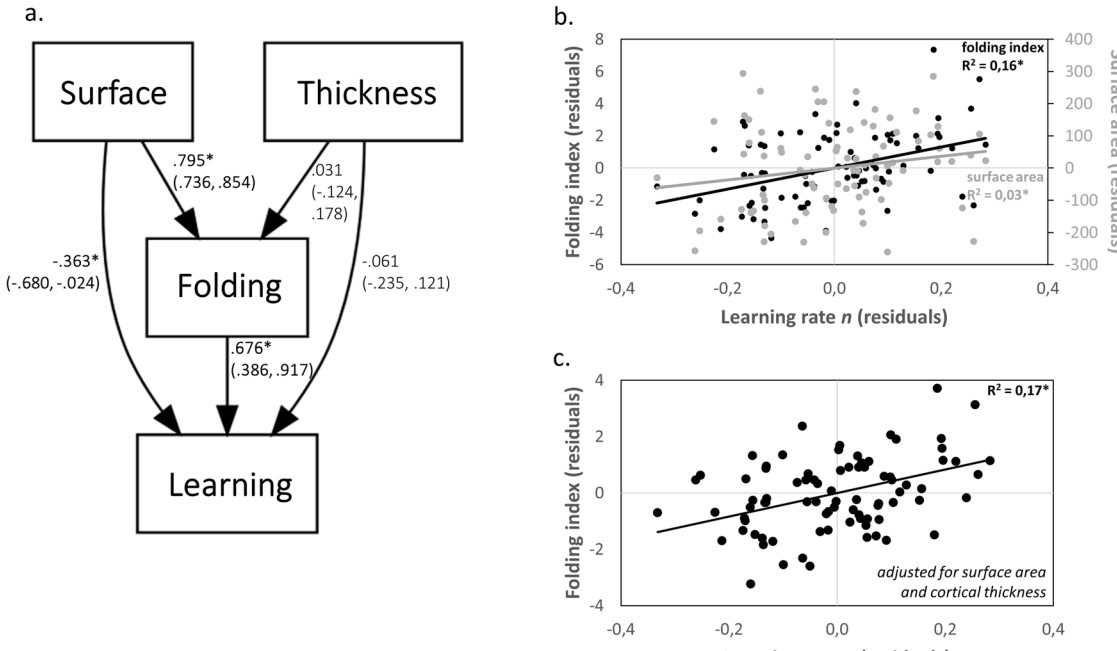

**Fig. 3 | Cortical surface area, but not cortical thickness, is related to the effect of cortical folding on learning.** Interrelationship between folding, thickness and surface area. **a** SEM depicting the relationships between cortical folding ('folding'), cortical surface area ('surface'), cortical thickness ('thickness'), and learning rate $n$ ('learning') in the left caudal superior frontal gyrus ROI. Standardized coefficients with 95% bootstrapped CIs are represented on paths. Correlations between average folding index and surface area in the ROI with learning rate $n$. Folding index is either unadjusted (**b**) or adjusted (**c**) for differences in surface area and cortical thickness. Note that all variables used in the model and for correlation analyses were corrected for differences in age, gender, height, study, head coil, baseline performance, and total intracranial volume. * indicate significant paths/correlations at $p < 0.05$ (with CIs not including zero).

**Fig. 4 | Cortical folding ties to learning rates independent of cortical myelination and cortical neurite density.** Analysis of microstructural tissue properties of the premotor cortex. Distribution of myelin-sensitive magnetization transfer saturation (MT) values (**a**) and the neurite density index NDI (**b**) across the left hemisphere. Color bars show regions of high MT or NDI in red (e.g., primary motor and somatosensory cortices) and regions of lower MT and NDI in blue (e.g., anterior prefrontal regions). Note the MT product-sequence-specific representation of MT values with a factor of 2. **c** MT and NDI values were analyzed in pre-SMA/SMA, the cluster in which cortical folding positively correlated with learning rate $n$ (Fig. 2a). **d** Pearson correlations between MT in superficial gray matter (GM), deep GM, and cortex-adjacent white matter with chronological age. **e** Pearson correlations between MT in superficial GM, deep GM, and cortex-adjacent white matter with learning rate $n$. **f** Partial correlations between cortical folding and learning rate adjusted for MT in superficial GM, deep GM, and cortex-adjacent white matter. * indicate significant correlations at $p < 0.05$, while ns indicates no significant correlation.

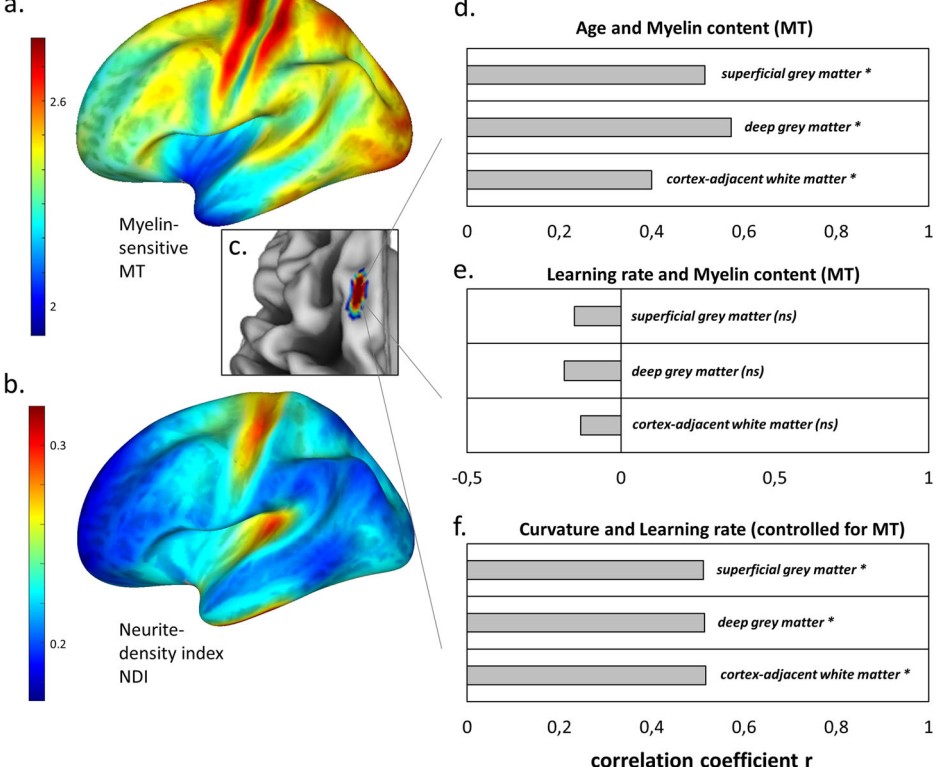

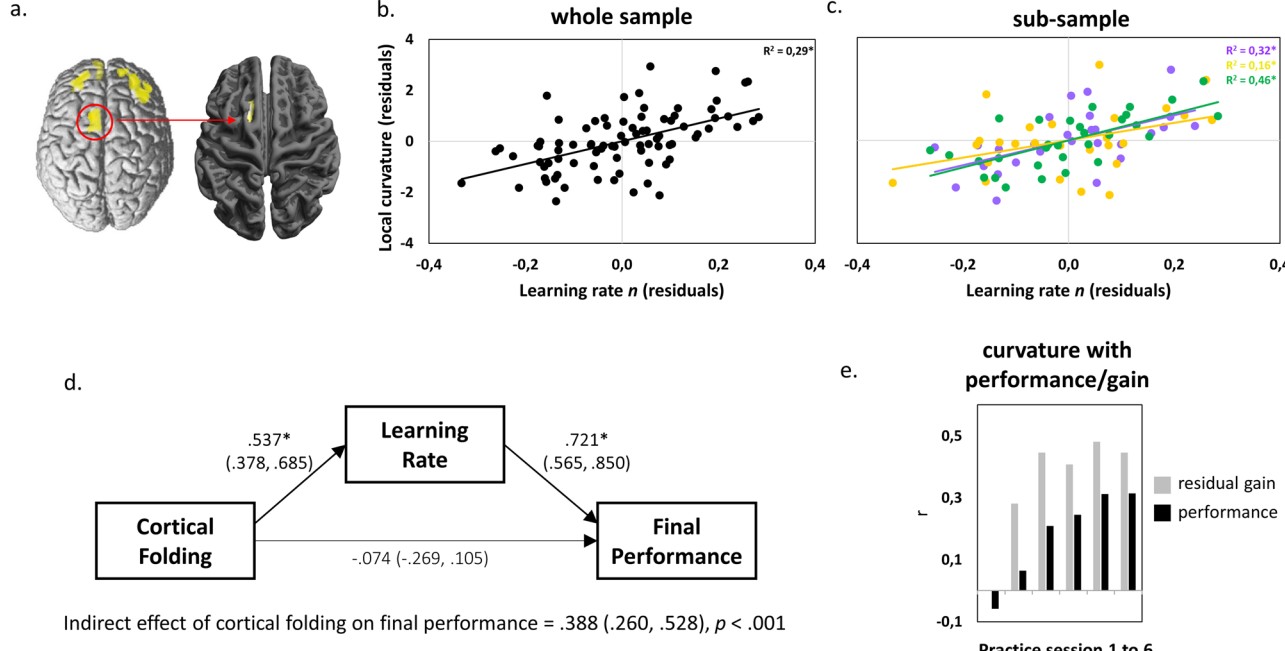

**Fig. 5 | Cortical folding is associated with learning in regions undergoing practice-induced structural plasticity.** Relationship between cortical folding and plasticity in the premotor cortex. **a** The clusters of significant learning-induced gray matter changes[43] (left) that overlapped with positive correlation of cortical curvature in pre-SMA/SMA and learning rate (right). **b, c** Whole-sample and sub-sample correlations between learning rate and cortical curvature in the pre-SMA/SMA cluster in A. **d** SEM depicting the relationship between cortical folding in pre-SMA/ SMA, learning rate (adjusted for $a$) and final performance on session 6. Standardized coefficients with 95% bootstrapped confidence intervals (CI) are represented on paths. **e** Pearson correlation coefficients between residualized cortical folding and motor performance. Gray bars represent session-specific performance controlled for initial performance in session 1 (i.e., residual gain) and black bars represent correlations with actual session-specific performance. * indicate significant correlations/ paths at $p < 0.05$ (with CIs not including zero).

(Fig. 2 and Supplementary Fig. 5). The relatively long practice period could have favored the identification of brain-behavioral relationships[34]. In addition, and compared to other studies testing for associations with individual differences in motor learning[34], our study included a relatively large sample size. This was possible by pooling datasets from three previous studies[43,48,58] with comparable MRI and behavioral learning data (Supplementary Tables 1–3). The cortical folding results from the main analysis ($N = 84$) were not dependent on the use of different T1-weighted MRI protocols within these sub-samples (Supplementary Table 2). Nevertheless, direct or close replication in new samples are essential to increase the level of evidence of our brain-behavioral model[66].

We found stronger associations between cortical folding and motor performance with increasing amounts of practice. This can be explained by the increasing impact of residual gains across practice (the improvement in performance from the first session to a later practice session) on absolute performance levels (Figs. 2e, 5e, Supplementary Fig. 6). In fact, performance gain fully mediated the effect of cortical folding on final performance (Figs. 2d and 5d). This suggests that the relation between cortical folding and (acquired) performance level may be an indirect consequence of cortical folding's relationship with learning ability (as suggested by our SEM's in Figs. 2 and 5).

Practice can further enhance individual differences in performance associated with relatively stable factors (e.g., aptitude, genotype, phenotypic, and other psychological traits), a view held in developmental psychology and behavioral genetics[39,67]. We interpret our finding as a reflection of interindividual differences in capabilities (rather than actual performance levels), mediated by the degree of cortical folding[35,36]. When our participants learned the postural task across six sessions, the impact of initial performance differences on subsequent performance gains decreased during practice (Supplementary Fig. 11). A large fraction of this decrease can be explained by variations in cortical folding of the pre-SMA/SMA, i.e., the influence of cortical folding on performance gains increases with practice. Future studies are required to disentangle the specific contributions of early

environmental factors to behaviorally meaningful variations in cortical folding.

A large network of cortical and sub-cortical regions is involved in gait and postural control[68], but MR image resolution limited our analyses to cerebral cortical associations with individual differences in balance learning. The supplementary motor area is critically involved in anticipatory postural control and gait[69,70]. This region also adapts its structure in response to balance learning[71]. Practice of the stabilometer balance task (as used in the present work) induces structural gray matter changes in the left pre-SMA/ SMA and microstructural changes in the underlying white matter tracts of the left centrum semiovale[43]. Practice-induced structural changes were also accompanied by increased functional connectivity between the pre-SMA/ SMA and medial parietal areas[72]. This indicates that postural learning is associated with the connectivity and folding pattern of the pre-SMA/SMA embedded within a wider cortical-subcortical network responsible for posture and gait control. Our study was not designed to disentangle potential contributions of (pre-SMA/SMA) cortical folding to a general learning ability essential for different types of motor or cognitive tasks. Future studies are required to test the general predictive ability of cortical folding by using different learning tasks within the same sample. The pattern of correlations identified in our study suggest that associations with cortical folding increase with longer practice periods.

Although the overall pattern of cortical folding is relatively stable across the lifespan, supportive interventions could have a significant impact on individual trajectories of motor learning[48]. In line with this, we found an overlap of meaningful folding variations with practice-induced plasticity in pre-SMA/SMA which is consistent with research using juggling as long-term motor learning paradigm[28]. A spatial overlap was found between juggling-induced gray matter changes in parietal regions and an association between baseline parietal gray matter volume with subsequent learning-induced performance improvements[28]. Together with our study, this supports future efforts to mitigate potential behavioral disadvantages related to cortical predispositions by using appropriate training methods. In this

context, augmentative interventions such as vigorous physical exercise can further improve learning in this particular postural task[48]. The beneficial effect of vigorous exercise on balance learning is mediated by structural and functional changes in the fronto-parietal brain network[48,73]. Thus, plasticity-inducing intervention strategies may be a fruitful approach to enhance learning beyond neural predispositions (see Supplementary Notes).

Lastly, cortical folding is the result of different mechanisms extrinsic and intrinsic to the cortical sheet. Extrinsic sources can be the volumetric constraints of the cranial vault harboring an expanded cortex or connected axons pulling cortical and sub-cortical regions closer together to enhance information transmission speed[74]. Intrinsic mechanisms can be a higher level of cortical neurogenesis, differential tangential expansion of upper cortical layers or neuropile growth[4,6]. Our structural equation model suggests that cortical folding statistically mediates the effect of higher cortical surface area on motor learning (Fig. 3a). In addition to cortical surface area, a partial correlation (Fig. 3c) indicates that surface area-independent extrinsic and/or intrinsic factors play a role in this folding effect. Ultimately, studies with ultra-high resolution MRI are required to reveal the microstructural sources of cortical folding that contribute beyond surface area (e.g., U-fibers, layer-specific microstructure).

NODDI and myelin-related quantitative MRI were the focus of our recent longitudinal training study[58] (this sub-sample is included in the main analysis of the current paper) as we wanted to examine plasticity using parameters with higher biological specificity. However, the multi-shell diffusion MRI and quantitative MRI sequences required to test these questions were not available or used in our previous longitudinal studies[43,48], so we analysed relationships between cortical folding, myelin and NODDI in participants from one subsample only ($n = 26$). The correlation between cortical folding in pre-SMA/SMA and learning rate was sufficiently high within this subsample (see Fig. 2c yellow). For this reason, we had the opportunity to test potential microstructural contributions (using NODDI and qMRI) to the folding-learning link using partial correlation analyses (Fig. 4).

Unfortunately, no motor transfer or retention tests were included throughout the longitudinal study designs. However, in one of our previous studies, retention performance was measured 3 months[43] and 15 months (not included in ref. 43) after the end of the intervention. The retention results (see Supplementary Fig. 1 in ref. 43) suggest that performance in this task is very stable. This would fulfill an important criterion of Schmidt and Lee's definition of motor learning as a relatively permanent change in a person's ability to perform a motor task/skill[75]. Furthermore, the trial-related behavioral data presented in Supplementary Fig. 12 and additional analyses of performance decrements between sessions (see legend of Supplementary Fig. 12) suggest a high level of retention across practice days.

We can only speculate about the reasons for the left-lateralized association between cortical curvature and motor learning rate (Fig. 2a). First, a meta-analytic study of fMRI BOLD signal changes suggests that the left dorsal premotor region is a hub for motor learning[76]. Second, our previous longitudinal training study demonstrated bilateral gray matter changes in this region within the first two weeks of balance training[43]. It is worth noting that the left-sided gray matter change appeared to be longer lasting than the right-sided effect, as we were only able to observe left-sided gray matter changes over the entire motor practice period (6 weeks). Based on our own results and other studies on bimanual coordination[77], we interpret these results to suggest that the left pre-SMA/SMA is a critical region in the coordination and learning of complex motor tasks. However, our study was not designed to distinguish between the role of left and right pre-SMA/SMA.

From an evolutionary perspective, advanced sulcal morphology in caudal frontal regions (rostral premotor areas) likely emerged after the common ancestor of humans and great apes split from that of other apes (e.g., gibbons) approx. 16 Mya, but before chimpanzees and humans diverged from their last common ancestor approximately 2.17 Mya[36]. Skeletal adaptations designate the evolution of orthograde bipedality in human ancestors likely around 4–7 million years ago[78]. Whether phylogenetic cortical brain adaptations, in addition to skeletal and vestibular organ

adaptations[79,80], contributed to bipedality and the efficient use of tools during bipedal stance and locomotion is currently unclear, but not unlikely in light of evolutionary expansions of associative frontal and parietal regions implicated in human mobility. Electrophysiological and clinical studies in humans show that the pre-SMA/SMA is critical for the predictive control of posture, e.g., during gait initiation and dynamic postural control[69,70,81,82]. Predictive postural control is required both for successful learning of new postural skills but also for efficient manual tool-use during upright stance and gait (e.g., to predictively counteract tool-use related shifts in the body's center-of-mass[83]). When our participants acquired a new postural skill on the stabilometer, motor control strategies shifted from an initial compensatory strategy (compensation of initially unpredictable board motion) to a predictive postural control strategy with anticipatory movements of arms, trunk and the upper body (i.e., the board motion becomes predictable and thereby controllable through anticipatory movements; see Supplemental Video files). Although characteristics of our postural learning task differ from the postural demands of our ancestors during terrestrial or arboreal stance and locomotion, the neural machinery of predictive postural control seems critical for successful behavior in both scenarios. Although speculative, the pre-SMA/SMA could have played an important part in meeting these higher demands, with a higher degree of cortical sheet size and folding predisposing for better capabilities to manage complex postural demands in arboreal and terrestrial contexts.

While the underlying factors of cortical folding are subject to intense research in the biological and physical sciences[84], our study investigated behavioral learning capacities associated with higher cortical folding levels in adult humans. The fact that the learning rates in our study were adjusted for differences in initial performance (and that cortical folding was also not related to initial performance differences) has implications for inclusive learning approaches. Individual learning capabilities, irrespective of initial performance conditions, may be associated with stable and region-specific morphological characteristics of the cortex. Under the assumption of physical constraints to the information processing capacity of the cerebral cortex[9], education seems critical for an individual to realize its potential in a particular domain regardless of their initial performance in that domain. Our study also showed that learning rates fully mediated the effect of cortical folding on asymptotic levels of performance at the end of practice. In that sense, improved human performance does not necessarily emerge from an extraordinary brain morphology, but rather from an interaction between fertile learning environments and remarkably high learning capabilities[36]. In our study with healthy human participants, high learning capability was partially reflected in the surface morphology of the human neocortex.

## Methods

### Experimental design
We analyzed magnetic resonance imaging (MRI) and behavioral data from three independent motor learning experiments involving adult human participants (see Participants). All participants with complete MRI and behavioral data from these studies were included in the analyses. MRI of the brain was performed before motor practice of a challenging new postural task on a stabilometer (see Postural task practice). Indices of motor performance and learning rate over several practice sessions (see Analysis of motor learning) were correlated with local indices of cortical folding from preprocessed MRI data (see MRI acquisition and MRI preprocessing). Statistical analyses involved vertex-wise comparisons of cortical curvature and region-of-interest (ROI) comparisons of cortical and sulcal morphology as well as intracortical microstructure (see Statistical analysis).

### Participants
Participant characteristics are depicted in Supplementary Table 4. For the main analysis, a sample of 84 right-handed participants with normal or corrected-to-normal vision (mean age of 23.5 years, age range of 19–35 years, 34 females, mean body height 174 cm, body height range 153–191 cm) was included from the datasets of three independent motor learning experiments[43,47–49]. In addition, data from ref. 85 was used to

increase the sample size for a separate analysis of short-term improvements in motor performance (only data from the first training session). All studies were performed in accordance with the Declaration of Helsinki and approved by the Ethics Committees of the Universities of Leipzig or Magdeburg (Germany). All ethical regulations relevant to human research participants were followed. Exclusion criteria were contraindications to MRI, body mass index (BMI) > 30 kg/cm$^2$, a history of neuropsychiatric diseases, left-handedness and prior experience with the task to be learnt. Individuals who were physically active for more than 4 h per week (including balance activities in other sports) were excluded from the studies. Only one of the five subgroups (drawn from the three longitudinal studies) included more physically active individuals (see Group 4 in Supplementary Fig. 7), who exercised more than 4 h per week. Here, our aim was to test whether people with high levels of physical activity have advantages in learning a balance-demanding task. This was not the case, which is why we included this subgroup in the main analysis. The correlations between cortical folding and learning rate per subgroup (Supplementary Fig. 7) indicate a comparable effect size in group 4 compared to the other subgroups with lower physical activity. All participants were screened for contraindications of MRI before participation. Participants were naive to the experimental setup and postural training procedure and were of comparable educational level (all participants had A-level).

## Postural task practice

Participants learned a challenging whole-body postural task on a stabilometer either on one practice session ($N = 131$) or over six practice sessions ($N = 84$). From the 84 participants, practice sessions were either distributed over six consecutive weeks with one training session per week ($N = 58$, study 1 and study 3) or distributed over four consecutive weeks with 1–2 practice sessions per week ($N = 26$, study 2). The stabilometer is a movable, seasaw-like platform attached to a superimposed pivot with a maximum board deviation of 26° to each tilt side (stability platform, model 16030 L, Lafayette Instrument). Participants were instructed to stand on the stabilometer board and hold/restabilize the platform within a tolerance interval of ± 3° from the horizontal (see Supplementary Video files). After each of the 15 trials (30 s in each trial) per practice session, participants received verbal feedback on motor performance, measured as the accumulated time (in seconds) that the Stabilometer board could be held within the target interval of ± 3° (time-in-balance). A short break of 2 min between trials was used to avoid fatigue. Each practice session lasted approx. 45 min while net practice time on the stabilometer was 7.5 min per session. To familiarize subjects with the task and to prevent falls, we allowed the use of a supporting hand rail in the first trial of session 1. Familiarization trials were excluded from the analysis. We used a discovery learning approach[86] in which no information about the performance strategy (only the trial-wise quantitative performance feedback) was provided during practice. Therefore, participants had to discover their optimal strategy to improve task performance (e.g., error correction strategy with legs, hip, and arms) based on by trial and error.

## Analysis of motor learning

The mean performance scores (mean of time-in-balance values across 15 trials) on each of the six practice sessions for each individual participant were fitted to a general power function, $y(x) = a * x^n$, which describes motor learning over longer timescales well[87]. In this function, the base $a$ denotes initial task performance, $x$ is training session (time devoted to practice), and the exponent $n$ indicates the slope of the function (rate of learning). Furthermore, early learning was calculated from performance data on the first practice session. For that, we subtracted the mean of the first five trials from the mean of the last five trials. We used learning rate ($n$), initial performance ($a$) and early learning (performance gain during session 1) as dependent variables in statistical analyses of brain-behavioral relationships. As expected from motor learning literature[52], initial performance negatively predicted learning rate (Fig. S1). To get an unbiased readout of learning ability, we adjusted $n$ for differences in $a$[53].

## Magnetic resonance imaging (MRI) acquisition

Anatomical T1-weighted Magnetization Prepared Rapid Acquisition with Gradient Echoes (MPRAGE) data[88] were acquired on a 3 T MAGNETOM MRI system (Siemens Healthcare, Erlangen, Germany) with 176 slices in sagittal orientation (study 1 $N = 27$: Tim Trio system using a 32-channel head coil, study 2 $N = 26$: Prisma system using a 64-channel head coil, study 3 $N = 31$: Prisma system using a 32-channel head coil). The imaging parameters used were as follows. Study 3: inversion time (TI) = 900 ms, repetition time (TR) = 2300 ms, echo time (TE) = 2.98 ms, flip angle = 9°, field-of-view (FOV) = 256 × 240 mm$^2$, spatial resolution = 1 × 1 × 1 mm$^3$; study 1: (TI) = 650 ms, (TR) = 1300 ms, (TE) = 3.46 ms, flip angle = 10°, (FOV) = 256 × 240 mm$^2$, spatial resolution = 1 × 1 × 1 mm$^3$; study 2: (TR) = 2600 ms; (TE) = 5.18 ms; flip angle = 7°; (FOV) = 256 × 256 mm$^2$; spatial resolution = 0.8 × 0.8 × 0.8 mm$^3$. Due to the potential influence of the radiofrequency head coil on brain morphometric indices[89] we corrected for this factor in the statistical models. In addition, we corrected for MRI scanner and MPRAGE sequence-specific effects using a separate nuisance covariate for each of the three studies.

MRI data sensitive to cortical microstructure were acquired in the context of study 2 (see above, $N = 26$) by a 3 T MAGNETOM Prisma system (Siemens Healthcare, Erlangen, Germany) using a 64-channel head coil. We acquired the multiparameter mapping (MPM) protocol[90] using three different predominant T1-, proton density (PD-), and magnetization-transfer (MT-)weighted images with multi-echo Fast Low-Angle Shot (FLASH) scans by appropriate choice of the repetition time (TR) and the flip angle α: TR/α = 23.0 ms/25° for T1w scan, 23.0 ms/5° for PDw scan, and 37.0 ms/7° for MTw scan. Multiple gradient echoes were acquired with alternating readout polarity at 8 equidistant echo times (TE) between 2.46 ms and 19.68 ms for T1w and PDw acquisitions and at 6 equidistant TE between 2.46 ms and 14.76 ms for MTw acquisition. Other acquisition parameters were: 0.8 mm isotropic resolution, 224 sagittal partitions, field of view (FOV) = 230 × 230 mm. The total acquisition time was 34.23 min. Transmit and receive field correction acquisition was done before every single image (56 sagittal partitions, field of view FOV = 230 × 230 mm, TR = 4,1 ms, TE = 1,98 ms).

Whole-brain diffusion-weighted (DW) images were obtained with a monopolar single-shot spin echo EPI sequence: TE = 74 ms; TR = 4970 ms; flip angle α = 90°; parallel Generalized autocalibrating partially parallel acquisitions (GRAPPA) acceleration factor = 2, matrix: 130 × 130; FOV = 208 × 208 mm$^2$; nominal spatial resolution = 1.6 × 1.6 × 1.6 mm$^3$; multiband acceleration factor = 2; phase-encoding direction: anterior » posterior; 228 isotropically distributed diffusion sensitization directions (38 at b = 1000 s/mm$^2$, 76 at b = 2000 s/mm$^2$, and 114 at b = 3000 s/mm$^2$) and 14 b = 0 s/mm$^2$ images (interleaved throughout the acquisition) were collected. The sampling scheme was designed according to Caruyer and co-workers (http://www.emmanuelcaruyer.com/q-space-sampling.php[91]). To generate appropriate fieldmaps to correct for susceptibility-induced distortions, nine b = 0 s/mm$^2$ images with reversed phase encoding (posterior » anterior) were also acquired. The total scan duration was 22 min 31 s.

## MRI preprocessing

**Data quality control**. MR images of all participants passed both the visual quality inspection and the CAT12 data quality checks. All scans from 131 participants reached a weighted average image quality rating (IQR) of 86.79% (range 80.64–89.87%) corresponding to a quality grade B while the long-term practice cohort ($N = 84$) reached a weighted average (IQR) of 87.32% (quality grade B; range 85.62–89.87%). In the main analysis ($N = 84$), data quality per sub-sample (Fig. 2) was comparable with slightly higher quality values for the T1-weighted images obtained with the 64-channel head coil (study2: 89.41%) than with the 32-channel head coil (study1: 86.41%, study3: 86.35%).

**Cortical curvature estimation**. T1-weighted images were preprocessed using the CAT12 toolbox, v12.7 r1738 (Christian Gaser, Structural Brain Mapping Group, Jena University Hospital; http://www.neuro.uni-jena.

de/cat12/[92]), within SPM12 v7771 (Statistical Parametric Mapping, Wellcome Trust Center for Neuroimaging; http://www.fil.ion.ucl.ac.uk/spm/software/spm12/) for Matlab R2017b (The MathWorks, Inc.). This image analysis pipeline allows for the computation of surface-based parameters based on, e.g., the mean curvature and procedures are described in detail on the CAT 12 website and manual (https://neuro-jena.github.io/cat/index.html#DOWNLOAD). All procedures followed the recommendations in the CAT 12 manual. Briefly, initial voxel-based processing involves spatially adaptive denoising, resampling, bias correction, affine registration and unified segmentation and provides starting estimates for subsequently refined image processing. Output images were then skull-stripped, parcellated into left and right hemisphere, cerebellum and subcortical areas as well as corrected for local intensity differences and adaptively segmented followed by spatial normalization. Subsequently, central cortical surfaces were reconstructed and topological defects were repaired using spherical harmonics. The refined central surface mesh provided the basis for extraction of local cortical folding metrics (e.g., local curvature) and resulting local values were projected onto each mesh node. Local gyrification[54] is revealed through estimations of "smoothed absolute mean curvature" based on averaging curvature values from each vertex of the surface mesh. Mean curvature is an extrinsic surface measure and represents change in direction of surface normals along the surface (normal are vectors pointing outwards perpendicular to the surface). Large negative values correspond to sulci and large positive values to gyri. The resulting values were averaged within a distance of 3 mm and converted to absolute values (both sulcal and gyral regions have positive values, see ref. 54). We then applied a surface-based heat kernel filter with Full-Width at Half Maximum (FWHM) = 20 mm, as recommended for vertex-wise gyrification in the CAT12 user manual. The resulting values give information about the local amount of gyrification. Finally, individual central surfaces were registered to the Freesurfer "FsAverage" template using spherical mapping with minimal distortions. Local gyrification values are transferred onto this FsAverage template.

**Reconstruction of cortical folding, surface area and thickness.** To assess local interactions of cortical folding, surface area and cortical thickness in the left caudal superior frontal gyrus and to manually define and label sulci in individual subjects native space, we additionally used FreeSurfer automated segmentation tools[93,94] (FreeSurfer 6) to reconstruct cortical surfaces (recon-all command; https://freesurfer.net/fswiki/recon-all) from all baseline T1-weighted MRI images of the long-term practice cohort ($N = 84$). Cortical reconstruction and volumetric segmentation were performed with the Freesurfer image analysis suite, which is documented and freely available for download online (http://surfer.nmr.mgh.harvard.edu/). The technical details of these procedures are described on the FreeSurfer website (https://surfer.nmr.mgh.harvard.edu/fswiki/FreeSurferMethodsCitation). Briefly, this processing includes motion correction of volumetric T1 weighted images, removal of non-brain tissue using a hybrid watershed/surface deformation procedure, automated Talairach transformation, segmentation of the subcortical white matter and deep gray matter volumetric structures (including hippocampus, amygdala, caudate, putamen, ventricles) intensity normalization, tessellation of the gray matter white matter boundary, automated topology correction, and surface deformation following intensity gradients to optimally place the gray/white and gray/cerebrospinal fluid borders at the location where the greatest shift in intensity defines the transition to the other tissue class. Once the cortical models are complete, a number of deformable procedures can be performed for further data processing and analysis including surface inflation, registration to a spherical atlas which is based on individual cortical folding patterns to match cortical geometry across subjects, parcellation of the cerebral cortex into units with respect to gyral and sulcal structure, and creation of a variety of surface-based data including maps of curvature and surface area. This method uses both intensity and continuity

information from the entire three-dimensional MR volume in segmentation and deformation procedures to produce representations of cortical thickness, calculated as the closest distance from the gray/white boundary to the gray/cerebro-spinal fluid (CSF) boundary at each vertex on the tessellated surface. The maps are created using spatial intensity gradients across tissue classes and are therefore not simply reliant on absolute signal intensity. The maps produced are not restricted to the voxel resolution of the original data thus are capable of detecting submillimeter differences between groups. Procedures for the measurement of cortical thickness have been validated against histological analysis and manual measurements.

Based on the group-level result of a correlation between motor learning ability and local cortical curvature in the left pre-SMA/SMA (Fig. 2a), we manually defined a region-of-interest (ROI) in the left caudal superior frontal gyrus (SFG, including pre-SMA/SMA) encompassing the cortex in SFG extending from the anterior edge of the superior precentral sulcus (joining the medial precentral sulcus) to the caudal part of the superior frontal sulcus (at the level of the gyral bridge between middle and superior frontal gyrus) and, in the medio-lateral dimension, the cortex running from the interhemispheric fissure to the superior frontal sulcus[95] on the Freesurfer "FsAverage" template brain. This ROI was projected to each participant's native space and local indices of cortical folding[96], cortical surface area and cortical thickness were extracted from the white matter surface (to avoid blood vessel contamination[8]) and averaged in this ROI.

**Local gyrification index (LGI).** We supplemented the analysis of local cortical geometry (curvature) with an analysis of a gyrification metric that depends on the ratio between the outer hull surface area and the local cortical surface area (called outer-surface-based gyrification indices). Therefore, we computed the local gyrification index[55] of freesurfer cortical reconstructions.

**MPM preprocessing.** The generation of (semi-) quantitative maps were performed using the hMRI toolbox (version 0.2.0, www.hmri.info[90]) using default parameters. Preprocessing steps are described in detail in ref. 49. Briefly, the hMRI toolbox uses approximations of the signal equations for small repetition time TR and small flip angles α, and estimates the longitudinal relaxation rate R1, the apparent signal amplitude A* map (proportional to the proton density map PD) and the MTsat. Here, we focus on myelin-sensitive magnetization transfer saturation ($MT_{sat}$) in pre-SMA/SMA. The $MT_{sat}$ was adjusted for T1 and B1 contributions, which often leads to additional variability[90]. The map was reoriented towards a standard pose by setting the anterior commissure at the origin and both anterior and posterior commissure (AC/PC) in the axial plane. This is a common step to increase the consistency in individual head positions prior to normalization and/or segmentation. The output resolution of this multi-parameter map was set to 1 mm isotropic.

**Neurite-orientation dispersion and density imaging (NODDI)—preprocessing.** In accordance with the majority of existing NODDI papers we opted for preprocessing tools provided by the FMRIB Software Library (FSL). Preprocessing steps are described in detail in ref. 97. After visual quality assessment, preprocessing of diffusion-weighted (DW) images started with the creation of a fieldmap using topup for later correction of susceptibility-induced distortions (unwarping). The approach combines the b = 0 s/mm² images acquired with reversed phase-encoding as described in the previous section. Using the eddy tool, data sets were corrected for susceptibility (using the fieldmap to emerge from topup), eddy current-induced distortions and head motion, and outlier slices were detected and corrected. Realignment of images in the course of motion correction was accompanied by appropriate correction of gradient directions.

NODDI parameter maps were estimated from corrected multishell DW images (b = 0 s/mm², b = 1000 s/mm², b = 2000 s/mm², and b = 3000 s/mm²) using the NODDI Matlab Toolbox v1.0.1 (http://nitrc.org/projects/

noddi_toolbox, default settings), implementing the model formulation of ref. 98. In brief, NODDI models the diffusion signal in each voxel as contribution from three compartments: intraneurite signal, referring to the space bounded by the membrane of neurites, extraneurite signal, referring to the space around the neurites (glial cells, cell bodies), and CSF signal, referring to the space occupied by CSF. In the mathematical formulation of the model, intraneurite signal is represented by a set of zero-radius sticks following a Watson distribution, extraneurite signal is represented by a cylindrically symmetric tensor and CSF is modeled as isotropic Gaussian diffusion. Here we focus on the microstructural neurite density index (NDI) map to emerge from the NODDI model, which is the fraction of tissue that comprises axons or dendrites.

### Statistics and reproducibility

Our main goals were to test for positive relationships between inter-individual differences in learning rate or motor performance with local cortical folding. In these analyses, we corrected for the influence of age, gender, body size, total intracranial volume (estimated using CAT12 module "Estimating TIV") and study (initial differences in $a$ were only adjusted in the analysis of learning rate).

**Motor behavior.** Short-term changes in motor performance (time-in-balance in seconds) in the first practice session ($N = 131$) were analyzed with repeated measures analysis of variance (RM-ANOVA) with within-subject factor TRIAL (15 levels) in SPSS (IBM SPSS Statistics, Version 28.0.1.0, Armonk, NY). Long-term changes in motor performance across the six practice sessions were analyzed with RM-ANOVA of the session mean values (mean of 15 trails per session) with within-subject factor SESSION (6 levels). Trial-to-trial variation in performance were calculated with the coefficient-of-variation (COV, standard deviation divided by the mean) for each session and subjected to RM-ANOVA with factor SESSION (6 levels). Session-specific inter-individual variation was quantified using interquartile range between the upper and lower 25% of mean performance values. Spearman correlations were used to relate mean performance values across sessions.

**Analysis of cortical folding on long-term learning, initial performance and short-term adaptation.** Our main predictions were tested with a multiple linear regression model in SPM12 with local cortical folding values across the cortex as dependent variable and learning rate $n$ ($N = 84$, corrected for individual differences in initial performance level $a$) or initial performance $a$ as well as short-term adaptation ($N = 84$ and $N = 131$) as predictors. In each analysis, we corrected for the influence of age[99], gender[100], body height[47], head coil[89], total intracranial volume[101] and training study[43,48,49]. Covariation between (nuisance) variables are shown in Fig. S2. Statistical inference of positive relationships between behavioral parameters and cortical curvature was performed across the whole cortex (exploratory analysis) with non-parametric permutation test (vertex-level T-statistics) and 5000 permutations. $p$-values were considered significant at an FWE corrected threshold of $p < 0.05$. Technical reproduction of significant effects was performed using a second MRI scan from the same participants. This second MRI scan was obtained after the motor practice period either six weeks (study 1 and study 3 from refs. 43,48) or four weeks (study 2 from ref. 58) after the baseline MRI scan. The cluster extent from the initial exploratory whole-cortex analysis (Fig. 2a) was used as inclusive mask and surface measures from the second time point were averaged in this respective mask. Cortical folding values in this mask were highly reliable across the two MRI time points ($r = 0.964$). The overlap between cortical folding and practice-induced plasticity in gray matter volume was calculated using a group-space mask of the cluster in pre-SMA/SMA where we previously identified gray matter changes across the 6-week practice period[43] (xyz MNI coordinate −12, 13, 64, cluster with highest $Z = 4.35$ across the whole brain). The voxel-space cluster (rendered brain see Fig. 2) was projected to the FsAverage surface template using

CAT12 surface tools. The cortical folding values in this mask as well as in the mask for technical replication were averaged and subjected to statistical analysis in SPSS. In correlation analyses, we used residualized learning rate and cortical curvature values (corrected for age, gender, initial performance, body height, head coil, TIV, training study) to determine reproducibility, effect sizes and coincidence of folding and plasticity (Figs. 2, 5 and S6).

In addition to the main cohort ($N = 84$) we included additional 47 participants from ref. 47 in the correlation of initial performance and short-term adaptation with cortical folding. These additional participants were measured on a Tim Trio MRI system using either 12-channel or 32-channel head coil (which was corrected for in the respective statistical model, for more details see ref. 47). Behavioral variables were either initial performance (mean performance of 15 trials in practice session 1) or early learning calculated as the difference between the mean of the last 5 trials and the mean of the first 5 trails from practice session one.

**Myelin-sensitive magnetization transfer saturation (MT) and estimates of neurite density index (NDI) from neurite-orientation-and-dispersion-imaging (NODDI) modeling of diffusion MRI.** Myelin-sensitive MT values were calculated from multiparametric quantitative MRI protocol[49] and NDI values were calculated from NODDI modeling of diffusion MRI data[97] within the gray matter in the study2-subsample ($N = 26$). Both MT and NODDI metrics are highly reliable[49,97] and calculation of NDI values within gray matter was adjusted according to ref. 102. Based on previous findings[1], MT values were extracted and averaged within three cortical depth-dependent tissue compartments (superficial and deep cortical gray matter [GM] and cortex-adjacent white matter) in individual space using CAT12 surface tools. For each compartment, a mean sampling function (average along surface normal) and a equi-distance mapping model with 7 steps was employed (start-point: superficial = −0.5, deep = 0, white matter = 0.5; endpoint: superficial = 0, deep = 0.5, white matter = 1.0). Superficial GM extends from the gray matter/CSF border to the central surface. Deep GM extends from the central surface to the gray/white matter border and the cortex-adjacent white matter extends from the gray/white matter border into the cortex-adjacent white matter. Due to lower resolution of diffusion data, NDI values were sampled from the whole GM compartment (start-point = −0.5, endpoint = 0.5). Resulting MT maps and NDI maps were resampled into template space and smoothed with filter size of 15 mm FWHM. To visualize MT/NDI distribution across the whole cortex (Fig. 4a, b), we additionally mapped and averaged MT values in the whole gray matter compartment (from gray matter/CSF to gray/white matter boundary). For statistical analysis, compartment-specific values were extracted from the region overlapping with the pre-SMA/SMA cluster (Fig. 2a), but also analyzed vertex-wise. We used residualized (corrected for age, gender, body height, TIV, initial performance) MT/NDI, learning rate, cortical folding and age parameters for all Pearson and partial correlation analyses or adjusted for these nuisance variables in SPM statistical models for vertex-wise analyses (except of the age by MT/NDI correlation in which we did not correct for age and initial performance).

**Structural equation modeling (SEM).** SEM was used to better understand the dependencies between motor behavior and cortical folding (Figs. 2, 3 and 5). For this purpose, we used the lavaan package[103] running in R (i386 4.1.1, R Core Team, 2020) and RStudio. In the first model (Fig. 2d), cortical folding in the pre-SMA/SMA and residualized learning rate $n$ were used as exogenous variables to predict final performance in practice session 6 (SEM fit indices $RMSEA = 0.000$, $SRMR = 0.000$, $CFI = 1.000$, $TLI = 1.000$). Note that values of cortical folding and final performance were not adjust for differences in initial performance in this analysis. In the second model (SEM fit indices $RMSEA = 0.000$, $SRMR = 0.000$, $CFI = 1.000$, $TLI = 1.000$, Fig. 5) we used the independent ROI in which practice-induced gray matter changes were found previously[43] (Fig. 2). In the third model (Fig. 3, SEM fit indices

*RMSEA* = 0.000, *SRMR* = 0.000, *CFI* = 1.000, *TLI* = 1.000), surface area, cortical thickness and cortical folding indices in the left caudal SFG were used as exogenous variables to predict learning rate *n*. All values were residualized for age, gender, body height, TIV, initial performance and training study with the exception that values of cortical folding and final performance in the first two models were not adjust for differences in initial performance. We calculated direct and indirect effects with 95% bootstrapped CIs using 5000 permutations.

## Reporting summary

Further information on research design is available in the Nature Portfolio Reporting Summary linked to this article.

## Data availability

All data are available in the main text or the supplementary materials. In addition, data used for the study/figures have been made freely available under https://doi.org/10.24352/UB.OVGU-2023-095. Requests for further information or MRI data should be directed to the corresponding author, M.T. (marco.taubert@ovgu.de).

## Code availability

The code used for the study/figures have been made freely available under https://doi.org/10.24352/UB.OVGU-2023-095. Requests for further information should be directed to the corresponding author, M.T. (marco.taubert@ovgu.de).

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

## Acknowledgements
We were funded by the German Research Foundation grant SFB 1436/C01 (MT, GZ).

## Author contributions
Conceptualization: M.T. and N.L.; Methodology: M.T. and G.Z.; Investigation and formal analysis, visualization, data curation, and writing—original draft: M.T.; Writing—review & editing: M.T., G.Z. and N.L.

## Funding

## Competing interests
The authors declare no competing interests.
