## [Peer Review File · Communications Biology]

Reviewers' comments:

Reviewer #1 (Remarks to the Author):

In the present manuscript, the authors have performed high-level research investigating the importance of cortical folding in our ability to learn a complex new motor skill. The authors have acquired T1-weighted (and myelin-sensitive and dMRI) data across multiple datasets to obtain a large sample size. The authors conclude that in a specific area in the brain, namely the left pre-SMA/SMA, cortical folding patterns (particularly driven by surface area) can predict long-term learning gains. Interestingly, an overlap in brain region was reported between the current findings and previously reported training-induced changes in GM volume. Overall, this is impressive work that surely provides new knowledge to the field. At the same time, I believe that the work could benefit from more structure and maybe a pre-selection of findings. It now feels as if too much information is shared. As a result the reader loses track of the main message that the authors would like to bring across.

I will list additional major and minor concerns below in a point-by-point matter:

- throughout the paper the authors should check inferences about causality or longitudinal changes. Sometimes claims are made or words are used that point to causality, where this does not appear to be justified
- In the introduction, there is a sudden jump from cross-sectional studies on cognition to motor learning. Are there any cross-sectional works on cortical folding motor performance, and/or are there learning-related studies on cortical folding in the cognitive domain? Suggestion to introduce the motor domain more clearly
- lines 102-108 > I would appreciate for a rationale for all of these additional analyses
- line 109: I suggest to rephrase this. the aptitude-treatment interaction is not directly tested in this work
- a strength of this work, i.e., large sample size, also raises some concerns. Authors report on different T1-W acquisition protocols with different head coils. It is reported that this information is included in the statistical analyses. I believe that the information that is shared is not sufficient to determine whether study differences induce unwanted effects. adding study as a covariate is not specific to the study protocol differences. can data be presented per study site with regards to the preprocessing steps. is data quality comparable across different studies?
- participant information on the sample of 84 participants is lacking (or I overlooked)
- participants are excluded based on prior experience/knowledge on the task. What about other elements, such as repetitive engagement in sports (e.g., gymnastics, soccer, etc).
- were participants with neurological diseases (including developmental disorders) excluded?
- what type of verbal feedback is provided? How was the time between 3 degrees calculated?
- it is mentioned that the practice duration is 45 min. However, actual practice time comes down to 7.5 minutes per session? or did I misinterpret. This seems to be a low amount of training
- is the first trial of the first session always discarded? or is it included in the initial performance calculation and the learning gain calculation?
- the MRI preprocessing section would benefit from multiple subheadings to increase readability
- Why did the authors choose to work with the curvature measure. Using the gyrification index seems to be much more accepted and used.
- Since the measures of cortical thickness and surface area are used only at baseline, why was this

analyses performed on the sample of 84 participants, and not on the full sample?

- I suggest that the authors mention explicitly why subcortical areas such as the cerebellum are not taken into account in this work. This seems to be a straightforward choice given the task design
- were there any falls / outliers in the behavioral data?
- Can the authors comment on not including a non-training group. maybe pure repetition of performing the task twice also correlates with cortical folding?
- The use of Pearson correlations valid? data from a normal distribution?
- p19: the term further MRI scan is not clear. What is tested here? is the same analyses performed on the data of baseline and post-training? In that case, i would like to see a direct comparison of folding patterns between these two time points.
- p19: I am afraid I cannot follow the reasoning behind the subgroup analyses. How were the subgroups determined and why?
- line 770: should this not be 14 trials?
- If the authors would like to keep the information on myelin and NODDI parameters, more information is needed with regards to the acquisition protocols and preprocessing pipelines. If I understood correctly, this comes from a small subsample. why? were exactly the same participants used for the cortical folding?
- Lines 36-38 in the abstract state something about the (in)dependence of cortical folding on cortical thickness, surface area and intracortical microstructural metrics. So, here these effects are reported as comparable effects, whereas different analyses are used: SEM for the one, and residualized correlations for the other. Please comment on this.
- is there any transfer - retention test included to determine actual learning?
- Figure 2F is not very informative. Consider revising
- line 161: using the term "predict" is risky. There could be a hidden variable that causes both.
- Figure 2D > would be helpful to show the effect between cortical folding and final performance without the mediator as well. Does this concern a full mediation?
- Fig 4F seems not very informative. How does it relate tot the correlative values without MT regressed out?
- I seriously wonder whether the analyses on the morphology of tertiary sulci is of added value here. I lost track here
- there is a clear lateralization effect that is not at all discussed. Why is the left pre-SMA/SMA so important?
- authors report about the overlap in the left pre-SMA/SMA across multiple studies and multiple modalities that are assessed. However, If I am not mistaken, no direct correlations are made. Are there no overlapping samples?
- line 410: long training duration. This is relative, as the actual training duration per session was rather short.

Reviewer #2 (Remarks to the Author):

This paper describes the association between local variation in gyrification within the SMA and motor learning. Throughout the manuscript and analyses, the authors attempt to isolate the association between local gyrification and initial motor capabilities independent of the rate of learning or

improvement in performance in the tasks. This is a positive aspect of the study and, overall, I am convinced by the data presented by the authors and believe this merits publication.

The authors also included analyses examining the frequency in variation of tertiary folds as well as the specific surface area and depth measures of these folds to their measures of motor learning; this was the weakest part of the study. The analyses are woefully underpowered and there are too few subjects in the different sulci variants groups to seriously address this question. Therefore, I would recommend that this aspect of the study be delayed. The paper is already too long and this would be an easy way to reduce the length of the manuscript without minimizing the impact.

I also had several other concerns about the analyses that I think merit some consideration by the authors.

First, while the authors controlled for total intracranial volume in their analyses, they didn't control for overall gyrification. I am guessing that the authors are assuming that the larger brains are more gyrified but that relationship is going to be far from perfect. In short, beside the inference made from the ROI analysis, how do the authors know that the associations they report do not reflect an overall more gyrified brain? Or Frontal lobe?

Second, when I read the title of the paper, I didn't imagine that the tasks they used assessed postural control which is a "motor" task but it certainly isn't once that pops to mind when thinking of motor learning. More importantly, the authors prescreened subjects for inclusion in the study that included factors like weight (for no obvious reason) and previous experience on related tasks. What would those be and how was this determined? Everyone walks and has to maintain balance every single day under a variety of circumstances.

On a related point, there was no "control" learning measure (i.e., no measures of motor or non-motor learning were included) as a means of assessing more general learning abilities. So how do the authors know that these effects are specific to postural learning, other than the references to previous work in other laboratories?

In the supplemental materials, there is a specific section devoted to evolutionary implications. This is basically entirely speculative which may be fine but why not include it in the body of the paper. Further, in light of the tasks used in this study, the introduction of an evolutionary argument should be framed within the context of the evolution of bipedalism and postural control.

There is substantial discussion about the interface between genes and environment on motor learning in both the introduction and discussion. None of this is terribly germane to the specific findings of this paper and could be eliminated or, at minimum, streamlined.

In summary, overall, I think this paper makes a meaningful contribution to the literature and merits publication with some minor revisions.

Reviewer #3 (Remarks to the Author):

The manuscript reveals that premotor cortical gyrification (but not cortical thickness or less cortical surface area) predicts long-term learning gains in a challenging motor task. The authors applied an innovative combination of advanced MRI data recording and analysis procedures as well as statistical methods. All methods and procedures are performed state-of-the-art and sufficiently well described. Results obtained support the conclusions of the paper. Herewith the paper provides a new marker for the prediction of (motor) learning outcomes and thus will have significant impact on the research field and probably for other dimensions of functioning and training as well. I only have one critical comment or question:

Given suspected close associations between other cortical structural and geometrical features (which are only partly confirmed in the later analysis) it might seem a bit arbitrary that particularly the folding pattern was selected as hypothesized predictor for learning gain. The authors could argue in more detail in the introduction to this point and why they decided in favor of gyrification and against surface area (both were revealed to depend on each other, in the end). Also, in the introduction (lines 86ff) the authors argue that "no study to date investigated whether neocortical folding, a relatively stable macroscopic property of the cortex, relates to motor learning capability". Is there potentially a reason for that and/ or isn't that true for the other structural/geometrical factors as well and thus not a sufficient justification? Do the authors want to argue that intraindividual differences in potential for motor learning and thus predictability is much bigger in mammalian species with folded as compared to those with unfolded cortex but also variations in surface area?

Reviewer #1 (Remarks to the Author):

Reviewer comments	Responses
1. In the present manuscript, the authors have performed high-level research investigating the importance of cortical folding in our ability to learn a complex new motor skill. The authors have acquired T1-weighted (and myelin-sensitive and dMRI) data across multiple datasets to obtain a large sample size. The authors conclude that in a specific area in the brain, namely the left pre-SMA/SMA, cortical folding patterns (particularly driven by surface area) can predict long-term learning gains. Interestingly, an overlap in brain region was reported between the current findings and previously reported training-induced changes in GM volume. Overall, this is impressive work that surely provides new knowledge to the field. At the same time, I believe that the work could benefit from more structure and maybe a pre-selection of findings. It now feels as if too much information is shared. As a result the reader loses track of the main message that the authors would like to bring across.	We thank the reviewer for the positive overall assessment of our work. We will explain below that we were able to structure the manuscript better and formulate it more clearly with the help of the reviewers' comments. We address the concerns in detail in the point-by-point answers below.
2. I will list additional major and minor concerns below in a point-by-point matter: - throughout the paper the authors should check inferences about causality or longitudinal changes. Sometimes claims are made or words are used that point to causality, where this does not appear to be justified	Thank you for this advice. We have removed all references to causality from our manuscript unless appropriately supported by empirical evidence. In the context of structural equation models (see for example Fig. 3), we explicitly refer to the term “statistical mediation” (instead of experimentally determined causality): line 409 “Our structural equation model suggests that cortical folding statistically mediates the effect of higher cortical surface area on motor learning (Fig. 3A).”
3. In the introduction, there is a sudden jump from cross-sectional studies on cognition to motor learning. Are there any cross-sectional works on cortical folding motor performance, and/or are there learning-related studies on cortical folding in the cognitive domain? Suggestion to introduce the motor domain more clearly	Based on your recommendation, we revised this part of the introduction and now include a short review of studies from the motor and cognitive domain. line 71 “A recent prospective observational study found strong correlations between cortical folding and intra-individual changes in cognition²⁷, although possible differences in the extent and intensity of practice could not be taken into account. In the motor domain, previous investigations revealed performance correlations with

	cortical folding (at a single point in time) in developmental and clinical samples ^{1,2}, as well as relationships between handedness with sulcation³, speech motor recovery with gyrification⁴ and expertise-related gyral differences in elderly musicians⁵. However, according to a recent review on individual difference predictors of motor learning⁶, the association between cortical folding and differences in motor learning remains unexplored.”
4. lines 102-108 > I would appreciate for a rationale for all of these additional analyses	Based on the reviewer’s suggestion, we now include a rationale for morphometric analyses in the introduction. line 102 “Cortical curvature can be examined in vivo using magnetic resonance imaging (MRI), providing a folding-related measure to investigate spatially-specific brain-behaviour relationships ^{7,8}. According to recent comparative⁹ and experimental¹⁰ studies, cortical folding appears to develop through differences in the size and thickness of the cortical sheet in interaction with intracortical microstructure (e.g., neuronal density). Therefore, indices of cortical surface area, cortical thickness and intracortical microstructure enable a complementary investigation of brain-behavioural associations of cortical folding. To comprehensively characterize the link between local cortical folding and motor learning, we pursue a stepwise analysis approach. Specifically, in cortical regions with learning-relevant geometrical features (cortical curvature), we further investigate contributions of cortical surface area, cortical thickness and intracortical microstructure (assessed using myelin-sensitive magnetization transfer saturation and neurite density index).”
5. line 109: I suggest to rephrase this. the aptitude-treatment interaction is not directly tested in this work	We have now removed the term “aptitude-treatment interaction” and reworded the phrase. line 111 “Using data sets from previous motor learning studies, we aim to disentangle the contribution of higher cortical folding either to superior (absolute) performance or adaptive capability (steeper learning rate above and beyond initial performance differences).”
6. a strength of this works, i.e., large sample size, also raises some concerns. Authors report on different T1-W acquisition protocols with different head coils. It is reported that this information is included in the statistical analyses. I believe that the information that is shared is not sufficient to determine whether study differences induce unwanted effects. adding study as a covariate is not specific to the study protocol differences. can data be presented per study site with	We conducted new analyses to show that the relationships between cortical folding and learning are not influenced by differences in the MRI measurement protocol (head coil, etc.). We are now providing this new information to readers in the supplement (Tables 2-3). We have also summarized detailed information about the measurement protocols in supplementary table 1. The results of the linear regression show that differences in the measurement protocols do not influence the strength of the identified relationship between folding and learning rate (Table S2). The results of the Chow test support this point and show that the regression lines of the subsamples with 32 and 64 channel coils do not differ. The regression lines are shown

regards to the preprocessing steps. is data quality comparable across different studies?	separately for the three subsamples in Figures 2, 5 and S6 and show that the slopes differ only minimally. In addition, data quality during pre-processing of T1-weighted MRI scans was obtained from CAT12 and reported in the methods section. Quality metrics suggest overall good-to-high quality of MRI data (quality grade B; range 85.62%-89.87%, see MRI preprocessing). In the main analysis (N=84), data quality per sub-sample (Fig. 2) was comparable with slightly higher quality values for the T1-weighted images obtained with the 64-channel head coil (study2: 89.41%) than with the 32-channel head coil (study1: 86.41%, study3: 86.35%). Overall, the analyses suggest that differences in MRI protocol did not influence our main outcome.
7. participant information on the sample of 84 participants is lacking (or I overlooked)	The participant information for the sample of 84 participants can be found in the legend of Figure 1: “(N=84, mean age 24.6 years, age range 19-35 years, 57 women, mean height 174 cm, height range 153-191 cm, all participants were right-handed)”
8. participants are excluded based on prior experience/knowledge on the task. What about other elements, such as repetitive engagement in sports (e.g., gymnastics, soccer, etc).	Individuals who were active for more than 4 hours per week (including balance activities in other sports) were excluded from the studies. Only one of the five subgroups (drawn from the three longitudinal studies) included more physically active individuals (see Group 4 in Supplementary Figure 7), who exercised more than 4 hours per week. We wanted to test whether people with high levels of physical activity have advantages in learning a balance-demanding task. This was not the case, which is why we included this subgroup in the main analysis. The correlations between cortical folding and learning rate per subgroup (Supplementary Fig. 7) indicate a comparable effect size in group 4 compared to the other subgroups with lower physical activity.
9. were participants with neurological diseases (including developmental disorders) excluded?	Yes. Based on our ethics approval, we excluded participants with (previous) neurological diseases. This was an exclusion criterion for participation in an MRI session during our balance training studies.
10. what type of verbal feedback is provided? How was the time between 3 degrees calculated?	We are sorry that we did not mention this clearly in the Methods section. We have expanded the relevant paragraph with this information: line 499 “After each of the 15 trials (30 seconds in each trial) per practice session, participants received verbal feedback on motor performance, measured as the accumulated time (in seconds) that the Stabilometer board could be held within the target interval of $\pm 3^\circ$ (time-in-balance).”
11. it is mentioned that the practice duration is 45 min. However, actual practice time comes down to 7.5 minutes per session? or did I misinterpret. This seems to be a low amount of training	Yes, the net training time is 7.5 minutes per session. We agree that this seems to be a low training amount in the context of classic (reactive) balance training, e.g. with different types of devices such as wobble boards, etc. However, the Stabilometer paradigm has previously been used in the literature as a valid model for motor learning

	^{11,12}. To draw conclusions about individual differences in motor learning, we focused on the comparatively short practice schedules per session, as described f. ex. in¹². Our original neuroimaging study¹³ was also designed to assess brain plasticity during the earlier period of skill acquisition, the time when most changes in performance were likely to occur. For this reason, we chose 6 practice sessions and a considerable time interval between sessions to allow the development of structural plasticity. Our subsequent studies using the stabilometer task were based on this original idea. In addition, it must also be taken into account that the stabilometer task is a continuous movement task without a recognizable beginning and end of the task-specific movement pattern as compared to discrete or serial movement tasks such as throwing or playing a piano composition.
12. is the first trial of the first session always discarded? or is it included in the initial performance calculation and the learning gain calculation?	No, only the first trial of the first session was discarded and not included in the calculation of the initial performance or learning gain parameters. The regular 15 experimental trials only began after this familiarization trial (with permission to grab the handrail) had been completed.
13. the MRI preprocessing section would benefit from multiple subheadings to increase readability	Thank you for this hint. We have added appropriate subheadings to this section.
14. Why did the authors choose to work with the curvature measure. Using the gyrification index seems to be much more accepted and used.	Gyrification and curvature parameters provide partially related but rather different information about cortical folding. While curvature is sensitive to local geometric differences of the cortical sheet (absolute curvature, saddle shapes, etc.), surface-based folding metrics such as local gyrification index (LGI) do not necessarily depend on local curvature but reflect differences in the size and ratio of how much cortex area is buried within the sulci (with varying definitions). For example, large fissures such as the Sylvian fissure have high LGI values due to the large enclosed cortical surface area, while smaller grooves such as tertiary or secondary sulci (where we have observed our plasticity effects in the past) do not. However, the latter sulci may bury cortex whose local curvature is much higher than in the larger fissures. As shown in the figure below, an individual's LGI is much more homogeneous throughout the cortex (top), while curvature is more sensitive to local geometry (bottom).  The figure displays six brain surface renderings arranged in two rows. The top row shows lateral, frontal, medial, and occipital views. The bottom row shows dorsal and ventral views. Each rendering is color-coded to represent either curvature (bottom row) or the local gyrification index (LGI) (top row). The curvature maps show high contrast, with red indicating high curvature and blue indicating low curvature, highlighting local geometric features like sulci and gyri. The LGI maps show a much more uniform, reddish-orange color across the entire cortical surface, indicating that the LGI is a more homogeneous measure of cortical folding compared to curvature.

The idea of our study was to use a folding metric that provides locally specific folding features in both gyral and sulcal areas and can be analysed within the SPM framework. A spatially unbiased metric makes it more likely to identify spatial overlaps between folding effects and our previous plasticity results (Fig. 5). Importantly, we supplemented the LGI analysis to provide readers with a comprehensive overview of the various folding features associated with learning. This analysis (see Supplementary Fig. 9) revealed positive correlations between learning rate and LGI in the caudal part of the left superior frontal gyrus.

15. Since the measures of cortical thickness and surface area are used only at baseline, why was this analyses performed on the sample of 84 participants, and not on the full sample?

The initial curvature analysis was used to locate cortical regions associated with learning rate or performance (see Introduction). We only had MRI data from 84 participants who learned the task in 6 sessions. The full sample included an additional 37 participants, but these participants only received one or two training sessions. Therefore, we could only use the full sample for analyses of initial performance and early skill acquisition effects. The main analysis of the longer practice period was based on the 84 participants. In this main analysis, we investigated the relationship between curvature and motor performance and learning throughout the cerebral cortex. We found that curvature in the left pre-SMA/SMA is positively related to the learning rate over 6 sessions. To obtain more detailed evidence about which structural properties influenced this cortical curvature or which structural properties influenced the covariance of cortical curvature with learning rate, we subsequently analysed the surface area and thickness values only in the left pre-SMA/SMA, i.e. only among the 84 participants with extended practice times. In contrast, the replication analysis focused on our main finding of cortical curvature. Here we wanted to determine possible time-dependent variations in the spatial extent and effect size of our main finding. For this reason, we performed the same curvature analysis across the entire cortex, but with another MRI scan of the same participants.

16. I suggest that the authors mention explicitly why subcortical areas such as the cerebellum are

We agree that subcortical regions such as the cerebellum play a fundamental role in postural control and learning. The literature is replete with studies linking the different

not taken into account in this work. This seems to be a straightforward choice given the task design	structures of the cerebellum or basal ganglia to motor learning and balance, and we have also previously observed training-induced structural changes in this balance learning paradigm¹³. With regard to cortical folding, the spatial resolution of our T1-weighted anatomical MRI scans (1 mm3 voxel size at 3T) does not allow us to fully resolve the anatomy of the cerebellum¹⁴. The limited spatial resolution in relation to the fine anatomy of the cerebellum, due to limitations in acquisition and analysis, has hampered in vivo anatomical characterizations and behavioural comparisons¹⁵. In contrast to the human neocortex, the surface of the cerebellum cannot be computationally reconstructed down to the level of individual small folds (folia) with conventional 3T MRI. Because a folium is typically only a few mm wide and the folia are so close together, it is difficult to completely resolve this due to partial volume effects. In the absence of high-resolution MRI data, we limited our analysis of cortical folding to the cerebral cortex. Given previous reports of differences in cerebellar volume in motor experts (e.g., ballet dancers, see Dordevic et al., 2018), possible associations between balance learning rate and cerebellar volume would be plausible, but this would be beyond the scope of our folding study. We now add the following sentence as justification: line 170 “Due to the relatively low initial image resolution (1x1x1mm voxel size), we had to limit our analysis of the folding to the cerebral cortex.” line 378 “A large network of cortical and sub-cortical regions is involved in gait and postural control¹⁶, but MR image resolution limited our analyses to cerebral cortical associations with individual differences in balance learning.”
17. were there any falls / outliers in the behavioral data?	There were no task-related falls in any of our participants who trained in this balance task. We did not collect fall-related data outside of the training context because we initially focused on young, healthy participants and we did not expect potential motor carryover effects from learning this balance task to real-world postural challenges in this age group. Figure 1 in the manuscript shows experimental behavioural data for participants who completed the training program. Our data show that each participant improves their performance during training, but to different degrees. No dataset was excluded due to outliers in performance values. By setting a threshold based on two standard deviations below and above the mean (the learning rate), two participants were in the outlier range. However, the relationship between learning rate and cortical folding in pre-SMA/SMA remained highly significant when excluding these two participants ($R^2 = 0.31$ for $n = 84$ and $R^2 = 0.29$ for $n = 82$).
18. Can the authors comment on not including a non-training group.	Our own initial studies using this balance paradigm (and from which we used data for this manuscript) were

maybe pure repetition of performing the task twice also correlates with cortical folding?

designed to examine brain plasticity in response to motor training and associations between behavioural improvements and plasticity¹³. Here, brain structure indices were our primary outcome measures, following, among other things, the methodology of the first plasticity studies by Draganski et al.¹⁷ and Scholz et al.¹⁸. For this reason, we did not include a control group with behavioural task-specific pre- and post-measurements at this time (but rather a passive control group with pre-post MRI scans). However, there are reasons to believe that cortical folding associations with learning cannot be explained by simply repeating the task twice: First, in addition to the main analysis of learning rates over 6 practice sessions, we also correlated cortical folding with individual differences in initial performance and the change in performance during the first training session. The results (Supplementary Figs. 3 and 4) show that cortical folding was not correlated with any of these behavioural parameters.

Second, we additionally conducted a control analysis in which we correlated cortical folding values in our ROI (pre-SMA/SMA) with trial-by-trial performance changes during the first session (changes from Trial 1 to Trials 2–15) [t2_1, t3_1...] and changes from trial n to trial n+1 [t3_2, t4_3...]. The results are presented in the following correlation matrix. All variables entered into the correlation matrix were corrected for differences in age, sex, body size, TIV, study, and initial performance on the first trial (instead of the mean of the entire initial session, which was used for Fig. S3). In the first column we show the correlations with cortical folding, indicating a high positive association with the learning rate (residual_gain) and only low correlations with the initial trial-wise performance changes within the first session.

Third, we found a strong spatial overlap between balance training-induced structural plasticity in the left pre-SMA/SMA and the correlation between learning rate and cortical folding in the same cortical region (Fig. 5).

We interpret these results as evidence that cortical folding in adults is specifically related to learning gains observed over several weeks compared to short-term changes in

	performance. 19. The use of Pearson correlations valid? data from a normal distribution?	Following reviewer advice, we checked all variables used for Pearson correlation analyses for normality. In case of non-normality (determined using the Shapiro-Wilk test), we now present Spearman rank correlation coefficients instead of the Pearson correlation coefficient throughout the manuscript. This was the case when raw measures of age or initial performance were used (e.g. Supplementary Fig. 1).
20. p19: the term further MRI scan is not clear. What is tested here? is the same analyses performed on the data of baseline and post-training? In that case, i would like to see a direct comparison of folding patterns between these two time points.	Thank you for pointing out this ambiguity to us. Because we used behavioural and brain imaging data from three previous longitudinal studies^{13,19,20}, we have more than one MRI scan per participant available. In the current manuscript, we used the MRI scan before the start of balance training for the main cross-sectional analyses. To test the reproducibility of the relationship between cortical folding and learning rate, we additionally used another MRI scan of the same participants (Supplementary Fig. 6). Here we chose to use the longitudinal post-training MRI scan because this post-training MRI scan was performed consistently in all three subsamples. Therefore, the reviewer is correct that the same analyses were performed on the MRI data at the beginning and after training. According to the reviewer's suggestion, we now also compared the cortical folding values longitudinally between the baseline value and the time after training. To this end, we conducted three analyses. First, we performed a pre-post comparison of cortical curvature across the whole brain in CAT12. This analysis revealed no significant increase or decrease in cortical curvature across the entire cortex, even at a more lenient statistical threshold ($p < 0.001$ uncorrected at the vertex level, cluster extent > 0). Second, we averaged the cortical curvature values at baseline and after training in the

	pre-SMA/SMA ROI. We found no significant increase or decrease in average curvature within the ROI (paired samples t-test, $p = 0.11$). Third, we correlated change in cortical curvature with motor learning rate to identify potentially meaningful patterns of change in cortical curvature. However, again, we found no association between these metrics (Pearson $r = .06$). These additional results are consistent with the high reliability of the pre-SMA/SMA cortical folding values between the two scans (see Methods section, page 20 “The cortical folding values in this mask were highly reliable across the two MRI time points ($r = 0.964$)).
21. p19: I am afraid I cannot follow the reasoning behind the subgroup analyses. How were the subgroups determined and why?	The aim of the subgroup analysis was to identify potential moderators for the relationship between cortical folding in pre-SMA/SMA and learning rate. Here we examined the extent to which the folding-learning correlation varies within our sample ($n=84$). For this purpose, we divided our sample into different subgroups depending on person-specific (e.g., gender) and performance-specific (e.g., level of initial performance) characteristics and measured the corresponding correlations within each subgroup. However, because this is a supplementary analysis and the method for identifying potential moderators requires simple slopes interaction and not descriptive comparisons between correlation coefficients, we decided to remove this subgroup analysis from the manuscript.
22. line 770: should this not be 14 trials?	No, because we only tried to familiarize the participant with the stabilometer board at the beginning of the first training session. After this single familiarization trial in the first session, participants received 15 trials in the balance task. The behavioural analysis included all 15 practice trials, i.e. without the familiarization trial.
23. If the authors would like to keep the information on myelin and NODDI parameters, more information is needed with regards to the acquisition protocols and preprocessing pipelines. If I understood correctly, this comes from a small subsample. why? were exactly the same participants used for the cortical folding?	Thank you for pointing out this ambiguity to us. NODDI and myelin-related quantitative MRI were the focus of our recent longitudinal training study¹⁹ (this sub-sample is included in the main analysis) as we wanted to examine plasticity using parameters with higher biological specificity. However, the MRI sequences (multi-shell diffusion MRI, quantitative MRI) required to answer these questions were not available or used in our previous longitudinal studies^{13,20}, so we only collected information on cortical folding, myelin and NODDI from participants in one subsample ($n=26$). However, the correlation between cortical folding in pre-SMA/SMA and learning rate was sufficiently high within this subsample (see Fig. 2C yellow). For this reason, we had the opportunity to test possible microstructural contributions (using NODDI and qMRI) to the folding-learning link using partial correlation analyses (Fig. 4). Following the reviewer's suggestion, we now provide detailed additional information on the acquisition protocols and pre-processing steps in the respective method sections (line 542ff).

24. Lines 36-38 in the abstract state something about the (in)dependence of cortical folding on cortical thickness, surface area and intracortical microstructural metrics. So, here these effects are reported as comparable effects, whereas different analyses are used: SEM for the one, and residualized correlations for the other. Please comment on this.	The reviewer refers to the following sentence of the abstract: “The individual folding predisposition to motor learning was found to be independent of cortical thickness and several intracortical microstructural parameters, but dependent on greater cortical surface area.” This is the conclusion reached based on our findings from the Freesurfer-based analyses of cortical thickness and cortical surface area in the main sample (n=84) and the analysis of cortical microstructure (NODDI and MT) in the subsample (n=26). Due to the limited sample size of the subsample, we were unable to calculate SEMs to link microstructure to learning and cortical folding. Therefore, we used (partial) correlation analyses to investigate possible microstructural contributions to the relationship between cortical folding and learning rate (Fig. 4). In contrast, we analysed the contributions of macrostructural indices to cortical thickness and cortical surface area using data from the entire sample in an SEM (Fig. 3). The SEM allowed us to test (statistical) relationships between learning, cortical folding, and the other macrostructural indices, based on predictions from comparative neuroanatomy⁹. Here we observed a significant contribution of cortical surface area to the effect of cortical folding on learning rate, but no significant contribution of cortical thickness (Fig. 3). While this surface contribution is consistent with the literature, it is discussed that other connectivity-based mechanisms may additionally contribute to the folding-learning association²¹. For this reason, we also analysed the partial correlation between cortical folding and learning rate, controlling for differences in cortical surface area and thickness (Fig. 3C). Here we found that the correlation was still present, suggesting that not only differences in cortical surface area contribute to folding, but also other factors that we were unable to further identify in the present manuscript. The results of microstructural analysis in the subsample (n=26) suggest that cortical neurite density (NDI) or cortical myelin (MTsat) may not contribute. In our opinion, ultrahigh-resolution MRI appears to be more sensitive to potential additional contributors or mediators due to its sensitivity to intracortical structural composition and cortex-adjacent fiber bundles (U-fibers). However, this cannot be addressed in the present study.
25. is there any transfer - retention test included to determine actual learning?	Unfortunately, no transfer test is included in the study designs. However, in one of our previous studies, retention performance was measured at 3 months and at 15 months¹³. Here the results (see figure below) suggest that performance on this task is very stable. This would fulfil an important criterion of Schmidt and Lee's definition, which reflects motor learning as a relatively permanent change in a person's ability to perform a motor task/skill²². Below are the trial-related data, including the long-term retention test at 15 months of follow-up (these follow-up data were not

included in¹³).

Furthermore, the trial-related behavioural data presented in Fig. 1B and additional analysis of performance decrements between sessions (see below) suggest a high level of retention across practice days.

The bars in the figure above show the difference in our primary behavioural outcome measure (time in target zone in seconds) between the mean of the first two trials (session $n+1$) and the mean of the last two trials of the previous session (session n).

26. Figure 2F is not very informative. Consider revising

This figure represents the modelled learning rate per subject. We have removed this part of the figure as it is (highly) redundant with Figure 1B.

27. line 161: using the term "predict" is risky. There could be a hidden variable that causes both.

We thank the reviewer for this important point. We removed the term "predict" in line 161 and all other sections of the manuscript to avoid misinterpretation of our results.

28. Figure 2D > would be helpful to show the effect between cortical folding and final performance without the mediator as well. Does this concern a full mediation?

Please see Supplementary Figure 8 for the vertex-wise analysis of the relationships between cortical folding and final performance (practice session 6). Final performance showed a non-significant trend for association with cortical curvature in left pre-SMA/SMA (local maximum at $x=-15$, $y=20$, $z=62$, $T=4.40$, whole cortex analysis with FWE - correction, $p = .053$, nonparametric t-statistic with 5000 permutations). We calculated the Pearson correlation coefficient for cortical folding values in the left pre-SMA/SMA area (the ROI we used in the correlation analyses for Fig. 2) and final performance. There was a positive correlation with $r = 0.396$ ($R^2 = 0.157$), which is lower than

	the correlation with learning rate. Therefore, we tested whether learning rate mediated the relationship between cortical folding and final performance. The SEM with learning rate as a mediator (Fig. 2D) suggests full mediation with no significant direct effect ($\beta = -.033$) of cortical folding as a predictor and final performance as a dependent variable.
29. Fig 4F seems not very informative. How does it relate to the correlative values without MT regressed out?	In Figure 4F, our aim was to demonstrate possible contributions of myelin-sensitive MT (in different layers of the pre-SMA/SMA) to the relationship between cortical folding and learning rate. The results of the partial correlation analyses show that MT does not play a role in this context, as the effect size remains stable at $R^2 = 0.26$.
30. I seriously wonder whether the analyses on the morphology of tertiary sulci is of added value here. I lost track here	According to the suggestions of reviewers 1 and 2, we removed all parts related to the morphology of the tertiary sulci from the manuscript.
31. there is a clear lateralization effect that is not at all discussed. Why is the left pre-SMA/SMA so important?	We can only speculate about the reasons for this left-lateralized effect. First, a meta-analytic study of fMRI BOLD signal changes suggests that the left dorsal premotor region is a hub for motor learning²³. Second, our previous longitudinal training study demonstrated bilateral gray matter changes in this region within the first two weeks of balance training. It is noteworthy that the left-sided gray matter change appeared to be longer lasting than the right-sided effect, as we were only able to observe left-sided gray matter changes over the entire exercise period (6 weeks). Third, our current research (which we will present at ECSS in Glasgow later this year) suggests that the level of initial performance moderates the relationship between cortical folding in the right pre-SMA/SMA and learning rate. Thus, it appears that individual differences in cortical folding in the right pre-SMA/SMA are positively correlated with learning rate, but more so in individuals who start at a lower level of performance. In contrast, the current manuscript suggests that differences in left pre-SMA/SMA are related to learning rate, even after accounting for initial performance. Based on our own results and other studies on bimanual coordination²⁴, we interpret these results to suggest that the left pre-SMA/SMA is a critical region in the coordination and learning of complex motor tasks. The right pre-SMA/SMA appears to be more important for learning at lower motor performance levels, i.e. in the early practice phase or in participants with lower baseline performance. However, our study was not designed to distinguish between the role of left and right pre-SMA/SMA.
32. authors report about the overlap in the left pre-SMA/SMA across multiple studies and multiple modalities that are assessed. However, If I am not mistaken, no	The sample we used in the current study consists of three subsamples that we previously examined in separate studies with different research questions. These research questions centred around balance and exercise-induced brain plasticity. In some of these studies, and thus also in subsamples of the current larger sample, changes in the left

direct correlations are made. Are there no overlapping samples?

pre-SMA/SMA were identified. For example, in one of our previous studies¹³ (n=14), we found changes in gray matter volume and functional connectivity in the left pre-SMA/SMA. The pre-SMA/SMA was also observed in one of our recent training studies¹⁹ (n=26). Both data sets were part of the current sample (n=84). A comparison between cross-sectional measurements of cortical anatomy (e.g., cortical folding) and training-induced changes in cortical structure is planned, but is beyond the scope of the current manuscript. A challenge is to coordinate training protocols and different imaging markers between the studies (partly on different MRI scanners) in order to identify common patterns of plasticity, which can then be compared with the baseline values of e.g. cortical folding. However, we have not yet reached this point. In the meantime, however, we can correlate effects within subsamples. Our most recent study¹⁹ provides the largest subsample (n=26) and a plasticity index (orientation dispersion index, ODI) that changed with balance training and also correlated with inter-individual differences in learning rate. While both measures individually correlated significantly with learning rate ($r = 0.59$ for ODI change and $r = 0.51$ for cortical curvature), a preliminary correlation analysis suggests a low correlation between ODI change and cortical folding in pre- SMA/SMA ($r = .18$, see figure below). When we entered the two variables, change in ODI and cortical curvature, in a linear regression model predicting learning rate, the results show that both ODI change and cortical curvature represent two distinct mechanisms significantly predicting individual differences in learning rate ($\beta=.53$, $T=3.3$, $p=.004$ for ODI change; $\beta=.36$, $T=2.2$, $p=.036$ for curvature).

33. line 410: long training duration. This is relative, as the actual training duration per session was rather short.

Thank you for bringing this issue to our attention. We added the word “relative” to the sentence.

Reviewer #2 (Remarks to the Author):

Reviewer comments	Responses
1. This paper describes the association between local variation in gyrification within the SMA and motor learning. Throughout the manuscript and analyses, the authors attempt to isolate the association between local gyrification and initial motor capabilities independent of the rate of learning or improvement in performance in the tasks. This is a positive aspect of the study and, overall, I am convinced by the data presented by the authors and believe this merits publication.	We would like to thank the reviewer for the overall positive assessment on our work and will address the concerns point by point below.
2. The authors also included analyses examining the frequency in variation of tertiary folds as well as the specific surface area and depth measures of these folds to their measures of motor learning; this was the weakest part of the study. The analyses are woefully underpowered and there are too few subjects in the different sulci variants groups to seriously address this question. Therefore, I would recommend that this aspect of the study be delated. The paper is already too long and this would be an easy way to reduce the length of the manuscript without minimizing the impact.	According to the suggestions of reviewers 1 and 2, we removed all parts related to the morphology of the tertiary sulci from the manuscript.
3. I also had several other concerns about the analyses that I think merit some consideration by the authors. First, while the authors controlled for total intracranial volume in their analyses, they didn't control for overall gyrification. I am guessing that the authors are assuming that the larger brains are more gyrified but that relationship is going to be far from perfect. In short, beside the inference made from the ROI analysis, how the authors know that the associations they report do not reflect an overall more gyrified brain? Or Frontal lobe?	We would like to draw the reviewer's attention to the Supplementary text (lines 2-14). In this supplementary analysis, we re-examined the correlation between local cortical curvature in SMA/SMA and learning rate ($R^2 = 0.317$, $p < 0.001$, Fig. 2B, already corrected for other covariates including TIV), but this time differences in the gyrification of the entire cortex were also taken into account. To do this, we used the total folding index²⁵, which is a single number that summarizes the overall extent of folding on a surface. Although a higher total folding index was significantly correlated with a steeper learning rate ($R^2 = 0.05$, $p = 0.041$), the relationship between local cortical curvature in pre-SMA/SMA and learning rate was only slightly reduced when we controlled for the total folding index (partial $R^2 = 0.286$, $p < .001$). This was also the case when we controlled for differences in the folding index of the frontal lobe (partial $R^2 = 0.289$, $p <$

	0.001), the average curvature of the frontal lobe (partial $R^2 = 0.279$, $p < 0.001$), or the average curvature of the entire cortex (partial $R^2 = 0.252$, $p < 0.001$). As with the overall folding index, each of the three global parameters had small but significant correlations with learning rate ($R^2 = 0.07$, $p = 0.012$ for frontal lobe fold index; $R^2 = 0.05$, $p = 0.033$ for frontal lobe curvature; $R^2 = 0.11$, $p = .002$ for total curvature). These results suggest that the relationship between pre-SMA/SMA cortical folding and learning rate is region-specific and relatively independent of the cortical folding of the entire cortex or frontal lobe.
4. Second, when I read the title of the paper, I didn't imagine that the tasks they used assessed postural control which is a "motor" tasks but it certainly isn't once that pops to mind when thinking of motor learning. More importantly, the authors prescreened subjects for inclusion in the study that included factors like weight (for no obvious reason) and previous experience on related tasks. What would those be and how was this determined? Everyone walks and has to maintain balance every single day under a variety of circumstances.	Overall, we excluded participants with previous experience in balance sports because Davlin (2004)²⁶ showed a positive effect of expertise (competitive gymnasts) on balance performance on the stabilometer. Our aim was to keep the between-subject variance in initial motor performance as low as possible, as brain networks may be differently involved (and therefore plastic) depending on the level of expertise. To assess possible influences of specific previous experiences, we simply asked participants to report their current and previous levels of physical activity (average hours per week) and type of sport during screening. The study by Davlin (2004)²⁶ also showed an influence of body height and weight on performance on the stabilometer. To avoid a strong influence of anthropometry on our behavioral data and possibly also on brain changes, we tried to keep BMI within range. There was no other reason for excluding participants with high BMI.
5. On a related point, there was no "control" learning measure (i.e., .o measures of motor or non-motor learning were included) as a means of assessing more general learning abilities. So how to the others know that these effects are specific to postural learning, other than the references to previous work in other laboratories?	Thank you for bringing this important point to our attention. We will first clarify why we did not include learning control measures or groups in our study and then provide reasons why we believe that the associations with cortical folding in pre-SMA/SMA could be specific to postural learning in this balance task. Our own initial studies using this balance paradigm (and from which we used data for this manuscript) were designed to examine brain plasticity in response to motor training and associations between behavioural improvements and plasticity¹³. Here, brain structure indices were our primary outcome measures, primarily following the methodology of the first plasticity studies by Draganski et al.¹⁷ and Scholz et al.¹⁸. For this reason, at this point we did not include a control group with behavioural task-specific pre- and post-measurements (but rather a passive control group with MRI sessions only). However, we are confident that cortical folding in the left pre-SMA/SMA is specifically related to postural learning in this task for the following reasons: First, our previous studies using the same movement task revealed changes in the structure and function of exactly the same brain region (pre-SMA /SMA)^{13,27}. This also applies to cortical microstructural changes¹⁹. The latter also

correlate with individual differences in the learning rate¹⁹. Learning other complex motor tasks, e.g., juggling, also induces changes in parietal gray matter volume, and initial differences in gray matter volume in the same region are also associated with learning gains²⁸. Thus, it appears that task-specific networks exhibit both a plastic response to training and their structure can constrain the extent of training gains. These findings suggest that pre-SMA/SMA both respond plastically to learning and that their structural properties correlate with individual differences in learning.

Second, we conducted additional analyses to corroborate the specificity of learning over longer practice periods (weeks). In addition to the main analysis of learning rates over 6 practice sessions, we also correlated cortical folding with individual differences in initial performance and change in performance during the first practice session. The results (Supplemental Figs. 3 and 4) showed that cortical folding was not correlated with any of these behavioural parameters. Since the change in performance in the first practice session was calculated as the difference between the mean of the last five trials and the mean of the first five trials, we also carried out a control analysis in which we compared cortical folding values in our ROI with the trial-by-trial performance changes during the first session (changes from trial 1 to trial 2-15 [t2_1, t3_1...] and changes from trial n to trial n+1 [t3_2, t4_3...]). The results are presented in the following correlation matrix. All variables entered into the correlation matrix were corrected for differences in age, gender, height, TIV, study, and initial performance in the first trial. In the first column we show the correlations with cortical folding, indicating a high positive association with the learning rate (residual_gain) and only low correlations with the initial trial-wise performance changes within the

first session.

We interpret these results as evidence that cortical folding in adults is specifically related to learning gains observed over several weeks compared to short-term changes in performance. We cannot rule out that cortical folding in the pre-SMA/SMA is associated with longer-term learning gains in other tasks. For this reason, we limited our conclusion to learning/adaptability and not specifically to posture learning. Future research is needed to determine the generality of folding predictors of learning. To clarify this, we have added the following sentence to the discussion:

Line 389 “Our study was not designed to disentangle potential contributions of (pre-SMA/SMA) cortical folding to a general learning ability essential for different types of motor or cognitive tasks. Future studies are required to test the general predictive ability of cortical folding by using different learning tasks within the same sample. The pattern of correlations identified in our study suggest that associations with cortical folding increase with longer practice periods.”

6. In the supplemental materials, there is a specific section devoted to evolutionary implications. This is basically entirely speculative which may be fine but why not include it in the body of the paper. Further, in light of the tasks used in this study, the introduction of an evolutionary arguments should be framed within the context of the evolution of bipedalism and postural control.

Following the reviewer's suggestions, we now insert the paragraph on evolutionary implications (previously in the supplementary discussion) into the discussion section of the main manuscript and removed the parts discussing the morphology of the tertiary sulcus. This new section in the discussion reflects on the possible role of premotor cortical folding in the development of bipedalism and predictive postural control.

line 422ff: “From an evolutionary perspective, advanced sulcal morphology in caudal frontal regions (rostral premotor areas) likely emerged after the common ancestor

	of humans and great apes split from that of other apes (e.g. gibbons) approx. 16 Mya, but before chimpanzees and humans diverged from their last common ancestor approximately 2.17 Mya²⁹. Skeletal adaptations designate the evolution of orthograde bipedality in human ancestors likely around 4-7 million years ago³⁰. Whether phylogenetic cortical brain adaptations, in addition to skeletal and vestibular organ adaptations^{31,32}, contributed to bipedality and the efficient use of tools during bipedal stance and locomotion is currently unclear, but not unlikely in light of evolutionary expansions of associative frontal and parietal regions implicated in human mobility. Electrophysiological and clinical studies in humans show that the pre-SMA/SMA is critical for the predictive control of posture, e.g. during gait initiation and dynamic postural control³³⁻³⁶. Predictive postural control is required both for successful learning of new postural skills but also for efficient manual tool-use during upright stance and gait (e.g. to predictively counteract tool-use related shifts in the body's center-of-mass;³⁷). When our participants acquired a new postural skill on the stabilometer, motor control strategies shifted from an initial compensatory strategy (compensation of initially unpredictable board motion) to a predictive postural control strategy with anticipatory movements of arms, trunk and the upper body (i.e. the board motion becomes predictable and thereby controllable through anticipatory movements; see Supplemental Video files). Although characteristics of our postural learning task differ from the postural demands of our ancestors during terrestrial or arboreal stance and locomotion, the neural machinery of predictive postural control seems critical for successful behavior in both scenarios. Although speculative, the pre-SMA/SMA could have played an important part in meeting these higher demands, with a higher degree of cortical sheet size and folding (Fig. 3) predisposing for better capabilities to manage complex postural demands in arboreal and terrestrial contexts."
7. There is substantial discussion about the interface between genes and environment on motor learning in both the introduction and discussion. None of this is terribly germane to the specific findings of this paper and could be eliminated or, at minimum, streamlined.	As per the reviewer's suggestion, we removed the parts discussing genetic effects on motor learning and cortical folding and avoided references to genetic studies in the introduction. To support our hypothesis of cortical predispositions for learning gains, we needed to make reference to the behavioural genetic studies of Fox et al.³⁸ and Williams & Gross³⁹.
8. In summary, overall, I think this paper makes a meaningful contribution to the literature and merit publication with some minor revisions.	Thank you.

Reviewer #3 (Remarks to the Author):

Reviewer comments	Responses
1. The manuscript reveals that premotor cortical gyrification (but not cortical thickness or less cortical surface area) predicts long-term learning gains in a challenging motor task. The authors applied an innovative combination of advanced MRI data recording and analysis procedures as well as statistical methods. All methods and procedures are performed state-of-the-art and sufficiently well described. Results obtained support the conclusions of the paper. Herewith the paper provides a new marker for the prediction of (motor) learning outcomes and thus will have significant impact on the research field and probably for other dimensions of functioning and training as well. I only have one critical comment or question:	We would like to thank the reviewer for the overall positive summary of our study.
2. Given suspected close associations between other cortical structural and geometrical features (which are only partly confirmed in the later analysis) it might seem a bit arbitrary that particularly the folding pattern was selected as hypothesized predictor for learning gain. The authors could argue in more detail in the introduction to this point and why they decided in favor of gyrification and against surface area (both were revealed to depend on each other, in the end). Also, in the introduction (lines 86ff) the authors argue that "no study to date investigated whether neocortical folding, a relatively stable macroscopic property of the cortex, relates to motor learning capability". Is there potentially a reason for that and/ or isn't that true for the other structural/geometrical factors as well and thus not a sufficient justification? Do the authors want to argue that intraindividual differences in potential for motor	Thank you for pointing out the lack of clarity in the motivation of cortical folding compared to other anatomical measurements. First, from a historical perspective, gyrification reflects an attribute of cerebral cortex morphology⁴⁰ that has been (sometimes incorrectly) associated with higher behavioural performance (see morphological descriptions by early anatomists, Spitzka, 1907 or Weiner, 2018 Journal of the History of the Neurosciences). Thus, it seems important to test for potential associations of cortical folding with different aspects of behaviour. Second, compared to the overall folding pattern, which appears to develop very early prenatally and postnatally, the size of the cortical surface peaks around age 11–12 years and undergoes a threefold postnatal expansion⁴¹. Studies of cortical surface area and cortical folding could thus provide different information about how cortical properties affect developmentally sensitive aspects of behaviour. Furthermore, spatial specific effects of cortical folding (in pre-SMA/SMA) may be easier to localize using geometric features of the cortical sheet (e.g. curvature) rather than by analysing differences in the lateral extension of the cortical surface. Cortical curvature measures the curvature of the cortex at a point, reflecting the rate of change of the surface's direction. In areas of high curvature, such as the peaks (gyri) and troughs (sulci) of cortical folds, the cortical surface bends sharply. Cortical curvature is a more localized

learning and thus predictability is much bigger in mammalian species with folded as compared to those with unfolded cortex but also variations in surface area?

measure than cortical surface area and can provide detailed information about the morphology of specific cortical regions.

Lastly, estimating the cortical surface area also appears to be analytically more difficult (personal correspondence with Christian Gaser). Cortical curvature therefore appears to be a suitable and locally specific marker to investigate early developmental influences on motor learning in adults. As suggested by the reviewer and in previous studies⁹, cortical folding depends to some extent on the cortical surface area and therefore we also wanted to clarify their contribution in the later part of the manuscript.

Previous research primarily used volumetric indices for cross-sectional comparisons with (motor) behaviour (including our own previous studies⁴²). However, due to the above reasons and possibly stronger dependencies of VBM parameters on the choice of MRI scanner, head coil, etc.⁴³, we analysed differences in cortical folding here.

Following the reviewer's suggestion, we now give reasons in the introduction why we choose cortical curvature as a suitable parameter for our analysis:

Line 56ff “Compared to cortical folding, which develops very early in prenatal and postnatal periods⁴⁴, cortical surface area increases threefold in the postnatal period and peaks at 11–12 years of age⁴¹. It therefore seems plausible to assume that differences in cortical folding in adults may represent consequences of early developmental influences on behavior⁴⁴.”

Line 98: “In the human brain, local geometric features of the cortical surface (e.g. cortical curvature) appear to fundamentally constrain brain function⁴⁵. Cortical curvature can be examined in vivo using magnetic resonance imaging (MRI), providing a folding-related measure to investigate spatially-specific brain-behaviour relationships^{7,8}. According to recent comparative⁹ and experimental¹⁰ studies, cortical folding appears to develop through differences in the size and thickness of the cortical sheet in interaction with intracortical microstructure (e.g., neuronal density). Therefore, indices of cortical surface area, cortical thickness and intracortical microstructure enable a complementary investigation of brain-behavioural associations of cortical folding. To comprehensively characterize the link between local cortical folding and motor learning, we pursue a stepwise analysis approach. Specifically, in cortical regions with learning-relevant geometrical features (cortical curvature), we further investigate contributions of cortical surface area, cortical thickness and intracortical microstructure (assessed using myelin-sensitive magnetization transfer saturation and neurite density index).”

References

1. Vareilles, H. de *et al.* Shape variability of the central sulcus in the developing brain: A longitudinal descriptive and predictive study in preterm infants. *NeuroImage* **251**, 118837; 10.1016/j.neuroimage.2021.118837 (2022).
2. Yan, S. *et al.* Impaired topological properties of cortical morphological brain networks correlate with motor symptoms in Parkinson's disease. *Journal of neuroradiology = Journal de neuroradiologie*; 10.1016/j.neurad.2023.09.007 (2023).
3. Sun, Z. Y. *et al.* The effect of handedness on the shape of the central sulcus. *NeuroImage* **60**, 332–339; 10.1016/j.neuroimage.2011.12.050 (2012).
4. Garnett, E. O. *et al.* Anomalous morphology in left hemisphere motor and premotor cortex of children who stutter. *Brain* **141**, 2670–2684; 10.1093/brain/awy199 (2018).
5. Rus-Oswald, O. G. *et al.* Musicianship-Related Structural and Functional Cortical Features Are Preserved in Elderly Musicians. *Frontiers in aging neuroscience* **14**, 807971; 10.3389/fnagi.2022.807971 (2022).
6. Ranganathan, R., Cone, S. & Fox, B. Predicting individual differences in motor learning: A critical review. *Neuroscience and biobehavioral reviews* **141**, 104852; 10.1016/j.neubiorev.2022.104852 (2022).
7. Luders, E. *et al.* Mapping the Relationship between Cortical Convolution and Intelligence: Effects of Gender. *Cerebral Cortex* **18**, 2019–2026; 10.1093/cercor/bhm227 (2008).
8. Schmitt, S. *et al.* Associations of gestational age with gyrification and neurocognition in healthy adults. *Eur Arch Psychiatry Clin Neurosci*; 10.1007/s00406-022-01454-0 (2022).
9. Mota, B. & Herculano-Houzel, S. Cortical folding scales universally with surface area and thickness, not number of neurons. *Science* **349**, 74–77; 10.1126/science.aaa9101 (2015).
10. Llinares-Benadero, C. & Borrell, V. Deconstructing cortical folding: genetic, cellular and mechanical determinants. *Nat Rev Neurosci* **20**, 161–176; 10.1038/s41583-018-0112-2 (2019).
11. Ryan, E. D. Prerest and Postrest Performance on the Stabilometer as a Function of Distribution of Practice. *Research Quarterly. American Association for Health, Physical Education and Recreation* **36**, 197–204; 10.1080/10671188.1965.10614679 (1965).
12. Wulf, G., Weigelt, M., Poulter, D. & McNevin, N. Attentional focus on suprapostural tasks affects balance learning. *The Quarterly journal of experimental psychology. A, Human experimental psychology* **56**, 1191–1211; 10.1080/02724980343000062 (2003).
13. Taubert, M. *et al.* Dynamic Properties of Human Brain Structure: Learning-Related Changes in Cortical Areas and Associated Fiber Connections. *J. Neurosci.* **30**, 11670–11677; 10.1523/JNEUROSCI.2567-10.2010 (2010).
14. Priovoulos, N. & Bazin, P.-L. Methods for cerebellar imaging analysis. *Current Opinion in Behavioral Sciences* **54**, 101328; 10.1016/j.cobeha.2023.101328 (2023).
15. Sereno, M. I. *et al.* The human cerebellum has almost 80% of the surface area of the neocortex. *Proc. Natl. Acad. Sci. U.S.A.* **117**, 19538–19543; 10.1073/pnas.2002896117 (2020).
16. Snijders, A. H. *et al.* Physiology of freezing of gait. *Annals of neurology* **80**, 644–659; 10.1002/ana.24778 (2016).

17. Draganski, B. *et al.* Changes in grey matter induced by training. *Nature* **427**, 311–312; 10.1038/427311a (2004).
18. Scholz, J., Klein, M. C., Behrens, T. E. J. & Johansen-Berg, H. Training induces changes in white-matter architecture. *Nat Neurosci* **12**, 1370–1371; 10.1038/nn.2412 (2009).
19. Lehmann, N. *et al.* Changes in Cortical Microstructure of the Human Brain Resulting from Long-Term Motor Learning. *J. Neurosci.* **43**, 8637–8648; 10.1523/JNEUROSCI.0537-23.2023 (2023).
20. Lehmann, N., Villringer, A. & Taubert, M. Colocalized White Matter Plasticity and Increased Cerebral Blood Flow Mediate the Beneficial Effect of Cardiovascular Exercise on Long-Term Motor Learning. *J. Neurosci.* **40**, 2416–2429; 10.1523/JNEUROSCI.2310-19.2020 (2020).
21. van Essen, D. C. A tension-based theory of morphogenesis and compact wiring in the central nervous system. *Nature* **385**, 313–318; 10.1038/385313a0 (1997).
22. Schmidt, R. A. & Lee, T. D. *Motor control and learning. A behavioral emphasis*. 5th ed. (Human Kinetics, Champaign, IL, 2011).
23. Hardwick, R. M., Rottschy, C., Miall, R. C. & Eickhoff, S. B. A quantitative meta-analysis and review of motor learning in the human brain. *NeuroImage* **67**, 283–297; 10.1016/j.neuroimage.2012.11.020 (2013).
24. Verstraelen, S. *et al.* Dissociating the causal role of left and right dorsal premotor cortices in planning and executing bimanual movements - A neuro-navigated rTMS study. *Brain stimulation* **14**, 423–434; 10.1016/j.brs.2021.02.006 (2021).
25. van Essen, D. C. & Drury, H. A. Structural and Functional Analyses of Human Cerebral Cortex Using a Surface-Based Atlas. *J. Neurosci.* **17**, 7079–7102; 10.1523/JNEUROSCI.17-18-07079.1997 (1997).
26. Davlin, C. D. Dynamic Balance in High Level Athletes. *Percept Mot Skills* **98**, 1171–1176; 10.2466/pms.98.3c.1171-1176 (2004).
27. Taubert, M., Lohmann, G., Margulies, D. S., Villringer, A. & Ragert, P. Long-term effects of motor training on resting-state networks and underlying brain structure. *NeuroImage* **57**, 1492–1498; 10.1016/j.neuroimage.2011.05.078 (2011).
28. Sampaio-Baptista, C. *et al.* Gray matter volume is associated with rate of subsequent skill learning after a long term training intervention. *NeuroImage* **96**, 158–166; 10.1016/j.neuroimage.2014.03.056 (2014).
29. Kaas, J. H. & Herculano-Houzel, S. (eds.). *Evolution of nervous systems. A comprehensive reference*. 2nd ed. (Elsevier, Amsterdam [u.a.], 2017).
30. Böhme, M. *et al.* A new Miocene ape and locomotion in the ancestor of great apes and humans. *Nature* **575**, 489–493; 10.1038/s41586-019-1731-0 (2019).
31. Spoor, F., Wood, B. & Zonneveld, F. Implications of early hominid labyrinthine morphology for evolution of human bipedal locomotion. *Nature* **369**, 645–648; 10.1038/369645a0 (1994).
32. Bramble, D. M. & Lieberman, D. E. Endurance running and the evolution of Homo. *Nature* **432**, 345–352; 10.1038/nature03052 (2004).
33. Wada, Y. & Nishimura, Y. Isolated astasia in acute infarction of the supplementary-motor area. *Case Reports* **2010**, bcr0120102618-bcr0120102618; 10.1136/bcr.01.2010.2618 (2010).
34. Zwergal, A. *et al.* Aging of human supraspinal locomotor and postural control in fMRI. *Neurobiology of Aging* **33**, 1073–1084; 10.1016/j.neurobiolaging.2010.09.022 (2012).

35. Richard, A. *et al.* Contribution of the supplementary motor area and the cerebellum to the anticipatory postural adjustments and execution phases of human gait initiation. *Neuroscience* **358**, 181–189; 10.1016/j.neuroscience.2017.06.047 (2017).
36. Yada, T. & Kawasaki, T. Circumscribed supplementary motor area injury with gait apraxia including freezing of gait and shuffling gait: a case report. *Neurocase* **28**, 231–234; 10.1080/13554794.2022.2071628 (2022).
37. Massion, J. Postural Control Systems in Developmental Perspective. *Neuroscience & Biobehavioral Reviews* **22**, 465–472; 10.1016/S0149-7634(97)00031-6 (1998).
38. Fox, P. W., Hershberger, S. L. & Bouchard, T. J. Genetic and environmental contributions to the acquisition of a motor skill. *Nature* **384**, 356–358; 10.1038/384356a0 (1996).
39. Williams, L. R. T. & Gross, J. B. Heritability of Motor Skill. *Acta genet. med. gemellol.: twin res.* **29**, 127–136; 10.1017/S000156600008606 (1980).
40. ROTH, G. & DICKE, U. Evolution of the brain and intelligence. *Trends in Cognitive Sciences* **9**, 250–257; 10.1016/j.tics.2005.03.005 (2005).
41. Bethlehem, R. A. I. *et al.* Brain charts for the human lifespan. *Nature* **604**, 525–533; 10.1038/s41586-022-04554-y (2022).
42. Lehmann, N. *et al.* Interindividual differences in gray and white matter properties are associated with early complex motor skill acquisition. *Hum Brain Mapp* **40**, 4316–4330; 10.1002/hbm.24704 (2019).
43. Streitbürger, D.-P. *et al.* Impact of image acquisition on voxel-based-morphometry investigations of age-related structural brain changes. *NeuroImage* **87**, 170–182; 10.1016/j.neuroimage.2013.10.051 (2014).
44. Cachia, A. *et al.* Towards Deciphering the Fetal Foundation of Normal Cognition and Cognitive Symptoms From Sulcation of the Cortex. *Front. Neuroanat.* **15**; 10.3389/fnana.2021.712862 (2021).
45. Pang, J. C. *et al.* Geometric constraints on human brain function. *Nature*; 10.1038/s41586-023-06098-1 (2023).

REVIEWERS' COMMENTS:

Reviewer #1 (Remarks to the Author):

Thanks to the authors for revising the manuscript based on previous comments. Overall, I feel that the quality has improved.

Some concerns that remain:

- point 4 > I believe that this section could be further improved. For a naive reader, this is not yet sufficiently clear.
- point 6 > Thank you for including this
- point 7 > this is quite unusual. I would prefer to find this information in the methods description.
- point 8 > I suggest the authors to include this exclusion criteria in the "participants" section. Please refer to Suppl. figure 7 in the text
- point 9 > can you add this information to the "participants" section. Now, it only mentions neuropsychiatric disorders.
- point 11 > can you include this information in the text. Now, the text is confusing/misleading, as the practice duration is far from 45 minutes per session. SO, include the net training time
- point 13 > thank you. For readability, I suggest to write MPM in full in the subheading
- point 16 > thank you for the additional information
- point 17 > That is reassuring. Can you add this information to the main text? including the corresponding p values.
- point 18 > thank you for the clear explanation
- point 20 > OK
- point 21 > makes sense
- point 23 > OK. are limitations of NODDI also included in the limitations/discussion section?
- point 25 > I suggest to include this information in the main text
- point 28 > I think that this should be made more clear in the main manuscript.
- point 29 > it does not play a role, or there might be other reasons.. please be cautious with your statements
- point 30 > makes sense
- point 31 > although the study was not designed to distinguish between left and right (which makes sense) it is a potentially interesting finding that is worth discussing in the main manuscript. I suggest to do so, albeit being hypothetically and/or based on earlier evidence
- point 32 > I think it is valuable to include this information in the text. It can tell us something about the mechanisms. What could be a driving factor affecting the two (ODI and curvature) that are affected by balance training
-

-

Reviewer #2 (Remarks to the Author):

I am satisfied with the responses from the authors.

Reviewer #3 (Remarks to the Author):

I very much appreciate the response of the authors to my comments. My concerns have been fully acknowledged in the revision of the manuscript. I therefore can only recommend accepting the manuscript for publication if the other reviewers, who had more expertise in the technical details of the methods applied, are also satisfied.